# Propagation of temporal and rate signals in cultured multilayer networks

Jérémie Barral [1,2], Xiao-Jing Wang[1] & Alex D. Reyes[1]

Analyses of idealized feedforward networks suggest that several conditions have to be satisfied in order for activity to propagate faithfully across layers. Verifying these concepts experimentally has been difficult owing to the vast number of variables that must be controlled. Here, we cultured cortical neurons in a chamber with sequentially connected compartments, optogenetically stimulated individual neurons in the first layer with high spatiotemporal resolution, and then monitored the subthreshold and suprathreshold potentials in subsequent layers. Brief stimuli delivered to the first layer evoked a short-latency transient response followed by sustained activity. Rate signals, carried by the sustained component, propagated reliably through 4 layers, unlike idealized feedforward networks, which tended strongly towards synchrony. Moreover, temporal jitter in the stimulus was transformed into a rate code and transmitted to the last layer. This novel mode of propagation occurred in the balanced excitatory-inhibitory regime and is mediated by NMDA-mediated receptors and recurrent activity.

[1] Center for Neural Science, New York University, New York, NY, USA. [2]Present address: Institut de l'Audition, Institut Pasteur, Paris, France. Correspondence and requests for materials should be addressed to J.B. (email: jeremie.barral@gmail.com)

 1

nformation in the nervous system, encoded as action potentials, propagates within and across the many networks in the brain. A simple substrate for signal propagation is a feedforward network, such as the sequential brainstem nuclei in sensory pathways or the cortical layers through which thalamic inputs from layer IV propagates. What signals are transmitted through feedforward networks is still under debate. Information may either be represented as the average number of spikes per unit time (rate coding)[1,2] or by their precise timing (temporal coding)[3]. The nervous system may also utilize a combination of both strategies, suggesting a continuum between these two extreme coding schemes[4]. Indeed, in some systems, timing information is converted and then transmitted via a rate code[5–7].

The feedforward architecture places specific constraints on the type of signals that propagate. Analyses of idealized feedforward networks consisting of randomly connected excitatory neurons indicate that transmitted signals default to synchronous events[8–12]. Activating a sufficiently large number of neurons in the input layer within a narrow temporal window (termed pulse packets)[8,10] or long stimuli[4,9,13] caused neuronal firing to either become more synchronous in the subsequent layers or to dissipate. From a neural code perspective, the development of synchronous activity degrades rate signals[13–15].

Asynchronous activity and hence rate information can theoretically propagate under certain conditions. In simulations, sparse but strong synaptic coupling decreases synchrony by effectively reducing the number of presynaptic inputs shared by neurons in successive layers[12,16]. Nearly complete asynchrony was also achieved by introducing background synaptic noise[13,16] or by embedding feedforward networks with strong connections within a recurrent network[11,12,17,18]. Modulating the relative timing or balance between excitation and inhibition also provides a means for selectively propagating temporal or rate signals[19–21]. Whether these conditions are met under physiological conditions have not yet been verified experimentally.

Idealized feedforward networks, although conducive to rigorous theoretical treatment, makes several simplifying assumptions and cannot account fully for complex response properties of neurons. In particular, recurrent connections within cortex provide additional drive to neurons and transform their activities. In sensory cortices (visual: ref. [22]; auditory: refs. [23,24]; somatosensory: refs. [5,25]), transient stimuli often generate tonic firing that extend past the stimulus offset. This prolonged response is most evident in cortical areas involved with working memory where transient stimuli (cue) evoke firing responses (hold period) that are long lasting and in some cases persistent[26,27]. Moreover, instead of becoming more synchronous in successive layers, the firing pattern can actually become less transient and more

persistent in progressively higher order brain structures[5,6,28,29]. Modeling studies suggest that recurrent activity and/or NMDA-mediated synaptic transmission underlie the prolonged activity[30] and are supported by experimental work in sensory systems, which demonstrates the role of NMDA current in the development of the late response[31,32].

Because of the difficulties in accounting for controlling the many variables in vivo, reduced preparations are often used to elucidate the underlying cellular and synaptic properties[33–35]. Experiments in a network of neurons cultured in a long, narrow continuous track suggest that rate signals can propagate through relatively long distances[36]. However, how these results are related to the theories is difficult to assess: the $Ca^{2+}$ signals were too slow to measure temporal signals and does not allow monitoring of subthreshold potentials, the stimulus parameters could not be systematically varied, and the network was continuous rather than having discrete layers.

Here, we examined signal propagation in in vitro cultures of excitatory (E) and inhibitory (I) cortical neurons grown in a multilayer chamber. As shown previously, networks in culture retain the general synaptic connection architecture documented in vitro and excitatory–inhibitory balance. Importantly, the cultured networks naturally reproduce salient firing responses observed in vivo with no fine tuning of the experimental conditions, indicating that the results reflect general operating principles independent of detailed organization or cell types[37]. To examine signal propagation, excitatory neurons that expressed channelrhodopsin were individually stimulated with a specified spatio-temporal pattern and activity of neurons in subsequent layers documented with whole-cell or cell-attached recording. In contrast to theoretical predictions, we find that the evoked firing far outlasts the transient stimulus and can propagate rate information successfully. Moreover, the firing rate is modulated by jitter in the stimulus, suggesting that information about the temporal dispersion of the input is transformed to a rate code. Whole-cell recordings, pharmacological manipulations, and computer simulations indicate that signal transformation and propagation is mediated by a combination of NMDA-receptors and recurrent connections.

## Results

**Multilayer network in vitro.** To examine experimentally the conditions for propagation of activity across layers, we cultured cortical neurons in chamber with multiple compartments in series (Fig. 1a, b). This design demarcated the different stages and hence permitted a systematic examination of signal propagation that was not possible with cultures grown in a single chamber[38]

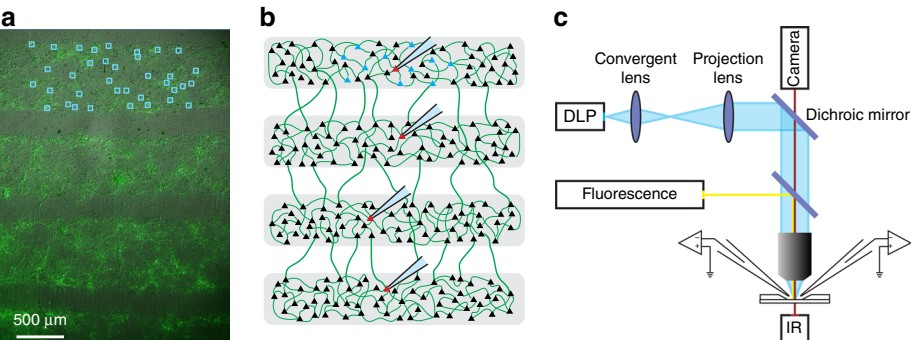

**Fig. 1** Stimulation of neurons in cultured multilayer networks. **a** Neurons, visualized with IR-DIC and fluorescent microscopy, are grown in distinct layers ($0.7 \times 6$ mm²) separated by 0.4 mm. **b** Schematic of the multilayer network. Cell-attached or whole-cell recordings could be performed from four neurons in sequential layers. **c** Using a Digital Light Processing (DLP) projector mounted on a microscope, brief light pulses (blue boxes in **a**) were delivered to neurons (in layer 1) that expressed ChR2 and a fluorescent tag (green)

or in a continuous track[34,36]. Importantly, optogenetic stimulation permitted independent stimulation of individual neurons with high temporal and spatial resolution (see below).

Although there were six layers, experiments were performed only in the middle four layers (henceforth designated as layers 1–4) to prevent edge effects due to fewer connections in the terminal layers (Supplementary Fig. 1). To confine cell bodies in layers, neurons were first cultured in rectangular compartments ($0.7 \times 6$ mm$^2$) separated by 0.4 mm spacers; after 24 h, the spacers were removed to allow bidirectional growth of axons. Some dendrites extended to the gap but are too small to reach the other layers[39]. At 14–21 days in vitro (DIV), neurons grew primarily in the compartments (henceforth termed layers) and formed recurrent and bidirectional synaptic connections within and across layers (see Supplementary Fig. 2). Previous studies showed that intrinsic and synaptic properties and the relative proportion and synaptic connection architecture between excitatory and inhibitory cells were similar to those measured in vitro[37,40].

To characterize the connectivity patterns between neurons, we performed paired whole-cell recordings and estimated the connection probabilities between cells. Neurons formed connections with other neurons in the same layer (henceforth termed recurrent connections) and with neurons in adjacent layers. The connection probability between neurons within a layer ($P_c = 0.3 \pm 0.06$; mean ± SEM; $n = 50$ tested connections in 13 networks; $533 \pm 132$ μm apart; mean ± SD) was comparable to those between neurons in two adjacent layers ($P_c = 0.23 \pm 0.04$, mean ± SEM; $n = 100$ tested connections in 13 networks; $666 \pm 140$ μm apart; mean ± SD). The connection probabilities were determined only by the distance between neurons and resembled the connection probability profiles of neurons grown in a single compartment (see Supplementary Fig. 2 and ref. [37]). Based on the connection profile and on the geometry, each neuron is estimated to connect to about 120 neurons in the same layer and to about 30 neurons in adjacent layers (Supplementary Fig. 1a, with a density of 300 neurons·mm$^{-2}$). Given the long distances between chambers (~1.1 mm, layer + gap), connections across non-adjacent layers were rare so that propagation occurs sequentially and did not "skip" layers (see simulations in Supplementary Fig. 1a). Both excitatory and inhibitory connections occurred across chambers; the connection architecture therefore resembles the layers of cortex more than the long-range excitatory connections between brain regions where inhibition is local.

**Propagation of activity in multilayer networks.** To examine propagation of activity across layers, we expressed channelrhodopsin (ChR2) in excitatory neurons using a transgenic line (see Methods). We optically stimulated neurons in the first layer using a computer-controlled Digital Light Processing (DLP) projector to deliver independent blue light pulses (Fig. 1c; see[37,41]) as follows. Approximately 20–40 neurons within a $0.7 \times 1.5$ mm$^2$ region were marked for stimulation with regions of interests (ROIs, Fig. 1a). A single brief blue light pulse (5 ms), which evoked reliable action potentials[37], was delivered to each neuron to evoke action potentials. The stimulus "packet" consisted of pulses delivered either synchronously or with temporal jitter. Extracellular spikes and intracellular membrane potential of neurons in layers 1–4 were recorded using cell-attached and whole-cell recordings, respectively.

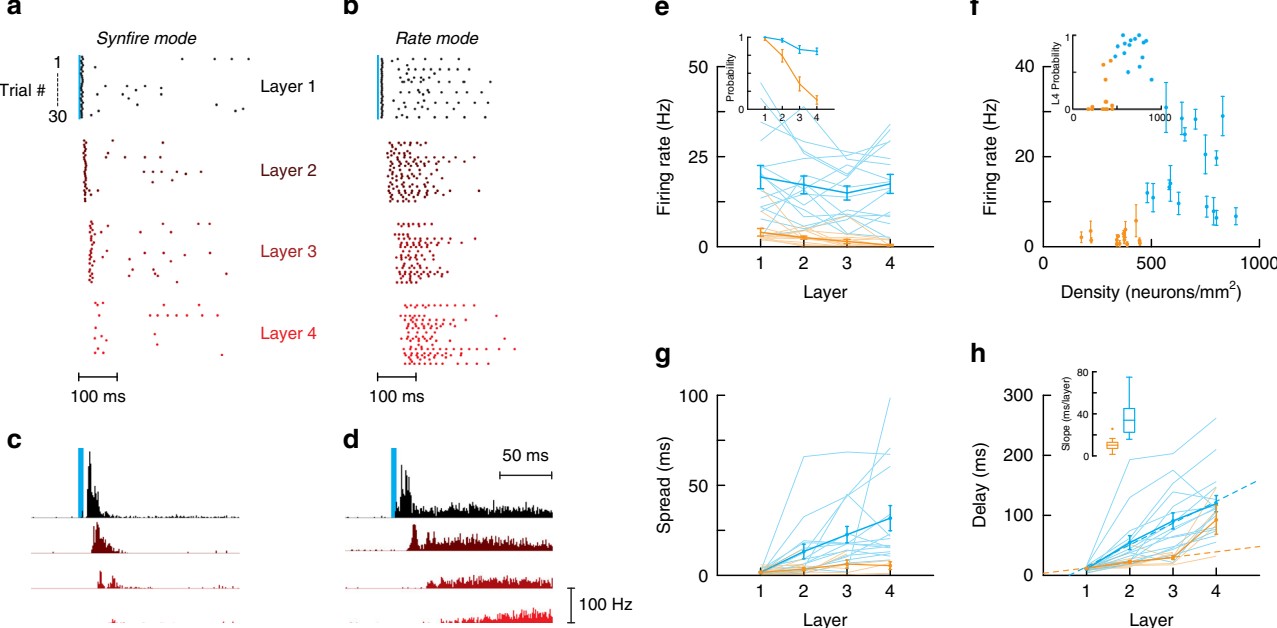

**Fig. 2** Activity propagation in multilayer networks in culture. **a, b** Examples of dot rasters showing spikes recorded in cell-attached mode from individual neurons in each layer. Synchronized light stimuli were delivered to ChR2-expressing (non-recorded) neurons in the first layer (30 repetitions of the light stimuli). Data in **a**, **b** are, respectively, from sparse and dense networks. **c, d** Average poststimulus time histogram (PSTH) of action potentials in each layer from sparse (**c**) and dense (**d**) networks. A single neuron was recorded in each layer (from black to red for layers 1–4, respectively). **e** Firing rate measured in a 300 ms time window after the stimulus. Inset, probability of evoking at least one spike vs layer in sparse (orange) and dense (blue) networks. **f** Firing rate averaged over all layers vs network density. Inset: spike probability in the 4th layer as a function of density. **g, h** Temporal spread (**g**) and delay (**h**) of first spike vs recorded layer in sparse (orange) and dense (blue) networks. In **h** dashed lines represent a linear fit to the data with slopes of 8.9 ms per layer and 36.2 ms per layer for sparse and dense networks respectively. The statistical significance between the two distributions of slopes for individual networks was assessed using Mann–Whitney U-test ($P = 1.7 \times 10^{-6}$). In **e–h** all data are shown and the mean ± SEM is represented as thick lines. Data for PSTH in **c**, **d** and plots in **e–h** are compiled from $n = 15$ sparse ($347 \pm 81$ neurons·mm$^{-2}$) and $n = 16$ dense ($686 \pm 122$ neurons·mm$^{-2}$) networks

There were two basic modes of propagation, depending on the density of the networks and stimulus parameters (Fig. 2a–d). In sparse networks (density $\lesssim$ 450 neurons·mm$^{-2}$), propagation occurred via what we henceforth term as synfire mode: light pulses delivered synchronously to 20–40 ROIs in the 1st layer evoked firing that was locked to the stimulus. Repeated delivery of identical stimuli evoked action potentials that exhibited little trial-to-trial jitter, as can be seen from the dot rasters of a single neuron (Fig. 2a) and from the sharp peaks in the poststimulus time histograms (PSTHs) compiled from several networks (Fig. 2c). However, the firing became less reliable and the activity disappeared in successive layers: the firing rate (Fig. 2e, orange; average in a 300 ms window after stimuli) and the probability of at least 1 spike occurring (inset) decreased to nearly 0 by the 4th layer. The jitter in first spike times, measured as the standard deviation of spike times across trials, remained small, with only a slight increase in the spread ($\lesssim$ 5 ms) in successive layers (Fig. 2g, orange), indicating that synchrony was preserved in the 2–3 layers that the activity propagated (see also below).

Propagation in the synfire mode was mediated primarily by unidirectional feedforward inputs despite the presence of reciprocal connections within and across layers. The spike onset increased linearly up to the 3$^{rd}$ layer with a slope of 8.9 ms per layer (Fig. 2h). The delay in action potentials between neurons across layers is composed of a conduction delay of 5 ms (conduction velocity[37,42] 200 µm·ms$^{-1}$ × 1.1 mm, layer + gap), a synaptic delay of 3 ms[37,43], and a delay due to the integration time associated with the neurons' membrane time constant (~20 ms[37]). Therefore, recurrent activity within a layer and feedback from downstream layers are unlikely to contribute substantially to the transient firing phase. Simulations with a network model that incorporated experimentally-measured synaptic potentials and architecture are consistent with predominant role of recurrent vs feedback connections (see below).

In denser networks, propagation occurred via what we henceforth term as a rate mode. Unlike the synfire mode, a stimulus packet delivered in the first layer evoked firing that had both a transient and a persistent phase (Fig. 2b, d). The transient component was similar to that in sparse networks: the trial-to-trial variability in the action potentials was small, as evident in a sharp peak in the PSTH. After a brief period of decreased activity, neurons fired over a period that often lasted several hundred milliseconds. The activity evoked in this persistent phase propagated reliably to layer 4, unlike the transient component, which mostly disappeared after layer 2. Transient activity propagated to layer 2 in 50% of the networks and to layer 3 in 12%, but never reached layer 4. Both the firing rate (Fig. 2e; blue) and probability of 1 action potential (inset) remained high across layers. This mode of propagation occurred reliably in networks with densities $\gtrsim$ 450 neurons·mm$^{-2}$ (Fig. 2f, blue). In contrast to the synfire mode, the temporal distribution (or spread) of the initial action potential increased substantially with layer (Fig. 2g). Moreover, the delay in the occurrences of the initial action potentials increased progressively at a rate of 36.2 ms per layer (Fig. 2h, blue). Because we based our analysis on the time of the first spike, the long delays indicate that propagation involved polysynaptic, recurrent connections within and possibly between layers.

These two modes of propagation could be simulated by networks of integrate-and-fire neurons with parameters derived from experimental measurements (see below). Varying the number of neurons and the synaptic strength according to the scaling rule measured experimentally[37], we found a transition between a synfire mode where propagation failed to a rate mode where activity displayed a transient and a persistent phase (Supplementary Fig. 3).

It is important to note that the light-evoked activities are not uncontrolled, all-or-none events but rather can be modulated by subtle changes in the stimuli such as the pulse rate of light stimuli (Supplementary Fig. 4) and the number of stimulated neurons (Supplementary Fig. 5). Moreover, as will be shown below, firing rate is also modulated by the temporal characteristics of the light stimuli and by the activity in layer 1.

**Propagation of pulse packets.** Propagation in feedforward networks is postulated to depend on the number and timing of active neurons in the first layer[8,10]. To test this hypothesis, we varied systematically the temporal distribution of the light pulses delivered to neurons in the first layer (Supplementary Fig. 6). The pulse packets were Gaussian distributed with standard deviations ($\sigma$) of 0, 5, 10, 15, and 20 ms (Fig. 3, left). The evolution of firing probability and spread (defined as the temporal distribution of the first spike) across layers could be tracked by constructing trajectories on a phase diagram (Fig. 3): the starting point of each trajectory (black arrows) reflects the spread of the light pulses in layer 1 and each segment thereafter corresponds to the values in layers 1–4.

Because we could only record simultaneously from a maximum of four neurons, the phase diagram could not be constructed by summing the responses of a population of neurons to a single stimulus;[10] instead, the phase diagram was constructed from the responses of single neurons in each layer to repeated, identical stimulation and the average compiled by pooling data from separate experiments (data from the 15 sparse and 16 dense networks in Fig. 2). Simulations that incorporated experimentally measured network variables[37] indicated that this method produced phase diagrams that were qualitatively similar to those obtained by combining population activities (Supplementary Fig. 7). Moreover, the distribution of spike times of four simultaneously recorded neurons in the same layer were strongly correlated with the distribution of spike times obtained by repeated stimulation of the same stimulus (Supplementary Fig. 8).

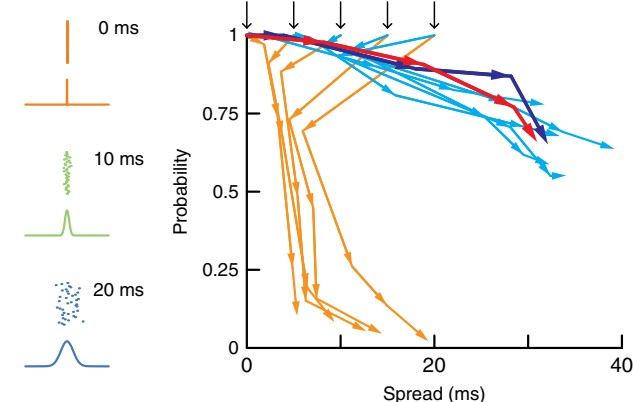

**Fig. 3** Phase diagram of spike propagation. Plot of spike probability vs spike spread for different initial conditions in sparse (orange) and in dense (cyan) networks. The jitter in the light pulse stimuli (pulse packet) delivered to the neurons was varied (left, examples of Gaussian distributed pulse packets with standard deviations of 0, 10, and 20 ms widths). Black arrows indicate stimulus jitters and arrowheads connected by lines indicate values in successive layers (from 1 to 4). Data are compiled from $n = 15$ sparse and $n = 16$ dense networks and are from the same networks in Fig. 2. Red and dark blue lines denote trajectories in sparse and dense networks, respectively, where all neurons in the field of view were activated synchronously (17 sparse networks of density = 303 ± 59 neurons·mm$^{-2}$ and 14 dense networks of density = 569 ± 80 neurons·mm$^{-2}$)

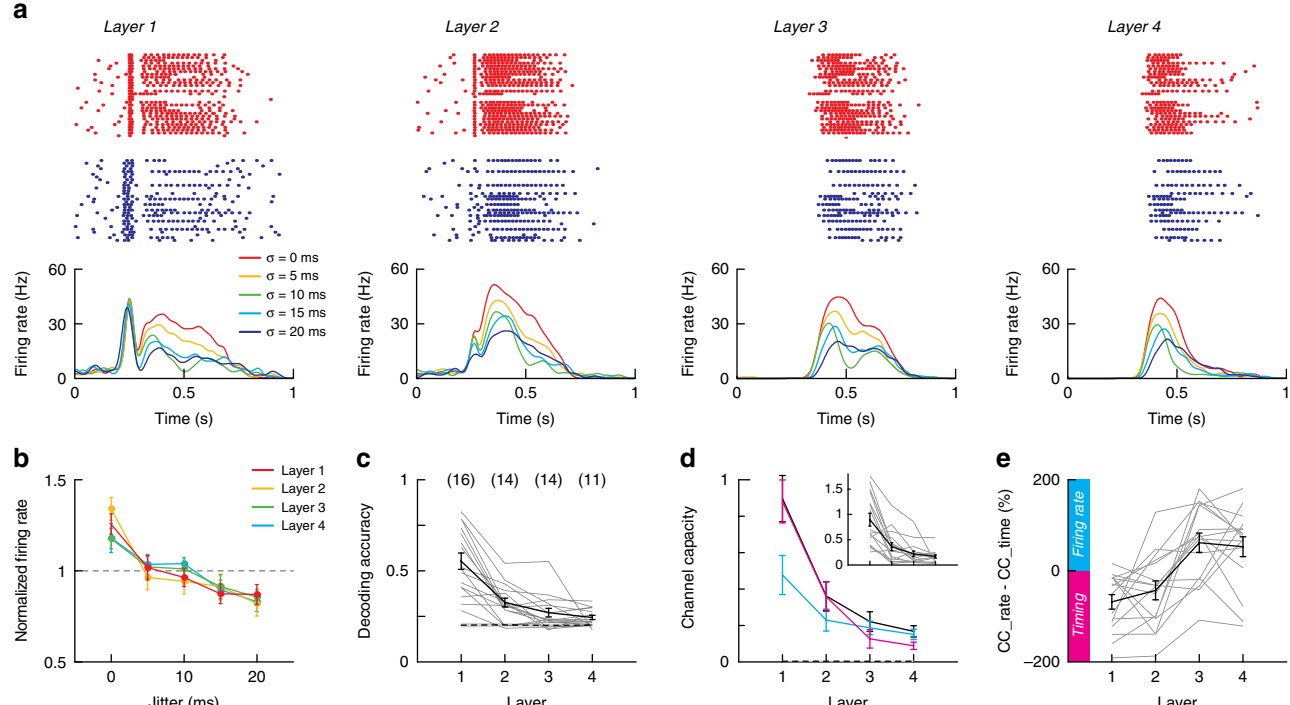

**Fig. 4** Propagation of information about the stimulus jitter in multilayer networks. **a** Top, dot rasters of spiking activity in individual cells in layers 1–4 (left to right) for stimulus jitters (σ) of 0 (red) and 20 ms (blue). Simultaneous cell-attached recordings were performed from one neuron in each layer. Bottom, PSTHs compiled from the spiking of neurons evoked with varying stimulus jitters (σ = 0, 5, 10,15, 20 ms; from red to dark blue). **b** Firing rate as a function of jitter for neurons in layers 1–4. The firing rate measured in a 300 ms time window following the stimulus was normalized (divided) by the rate averaged over the different jitters. **c** Decoding accuracy quantified as the probability of correctly classifying stimulus jitter in each layer. Decoding used time of the first spike and firing rate as predictors (see text). Numbers in brackets indicate the number of individual neurons (out of 16 neurons for each layer) with significant decoding accuracy with a *p*-value of 0.01 (see Methods). **d** Channel capacity computed using the time of the first spike and the firing rate (black), only the firing rate (magenta), or only the time of the first spike (cyan) as predictors. Inset: channel capacity for individual data. **e** Difference between the information carried by the firing rate and the information carried by the time of the first spike relative to their sum. Data are compiled from the same *n* = 16 dense networks of Fig. 2 and are presented as mean ± SEM. In **c**–**e** all data are shown as thin lines

Thus, to a first approximation the phase diagrams calculated from individual neurons resembles that calculated from a population.

The trajectories deviated from predictions of theory[10]. In the rate mode (Fig. 3, cyan), the trajectories starting at different initial stimulus conditions (here defined as layer 0; cf[37]) approached each other in the first layer (next segment) and then co-varied thereafter. The trajectories did not approach stable non-zero attractors with low jitter and high spiking probability but rather moved towards the direction of increasing spread. Finding potentials attractors would require more layers than present in the experiments.

The trajectories in the synfire mode (Fig. 3, orange) were almost perpendicular to those in the rate mode: all followed paths that clustered in the first layer regardless of the initial conditions and then moved towards zero probability with a relatively small change in the spread. Increasing the amplitude and decreasing the width of the pulse packet, which was accomplished by synchronously activating all ChR2-expressing neurons within the field of view (0.7 × 1.5 mm²) in layer 1, produced mixed results: all (17/17) showed a rate-like propagation as with the dense network but with some also propagating the transient component to layer 4 (8/17). Hence, the state-space trajectory did not simply move towards an attractor with increasingly narrow spread and high probability firing but rather acquired a rate mode that resulted in a widening with successive layers (red). Similar stimuli delivered to networks already exhibiting rate mode propagation did not change the trajectory (dark blue).

**Encoding and transmitting information about the stimulus jitter as rate.** The temporal information that is lost with the disappearance of the transient component (Fig. 2b, d) and the increase in spread of the first spike (Figs. 2g and 3) is transformed and propagated via a rate code. In the first layer, systematically increasing the stimulus pulse jitter (σ = 0, 5, 10, 15, 20 ms) caused progressive decreases in the neurons' firing rate evoked in the persistent phase (Fig. 4a, b; Supplementary Fig. 9). In the next layers, the transient component disappeared but the differences in firing rate remained.

The observation that the spikes were tightly distributed in the initial layers and subsequently replaced with a sustained component in deeper layers suggested that both timing and rate are important variables for propagation. To examine the evolution from a temporal to a rate code across layers, we trained a binary classification decision tree to decode the jitter in the stimulus pulses (see Methods). Using as predictors the firing rate in a 300 ms time window after the light stimulation and the time of the first spike, we computed the decoding accuracy (Fig. 4c, black) and the channel capacity (Fig. 4d, black). Decoding accuracy and channel capacity decreased in successive layers but remained above chance. Channel capacity represents the upper bound of the mutual information between the stimulus feature (i.e. the temporal spread σ) and the response (i.e. spike time and firing rate). In the first layer, the channel capacity averaged over 16 networks was ~0.9 bits, which represents about 40% of the upper bound defined as the total entropy of the stimulus (~2.3 bits). In subsequent layers, this measure decreased to 0.2 bits in

layer 4 but remained above chance level (statistically significant decoding accuracy was observed in 11/16 networks). There was some variability in channel capacity in individual experiments with some reaching values as high as 1.8 and 0.6 bits in layers 1 and 4, respectively (inset in Fig. 4d). In the synfire mode, decoding accuracy dropped to zero because activity did not propagate (Supplementary Fig. 10). In simulations, modulation of firing rate and propagation of information about the stimulus jitter were also found in dense but not in sparse networks (Supplementary Fig. 11).

The transient component, though dissipating in deeper layers, improved information transmission in the initial layers. To determine the relative contribution of timing and rate to channel capacity in successive layers, we trained the classifier solely with either first spike timing (Fig. 4d, magenta) or firing rate (blue). In layers 1 and 2, spike timing contributed more to channel capacity than firing rate (Fig. 4e); in layer 3–4, firing rate accounted for most of the channel capacity. Taken together, the results suggest that information about the temporal dispersion of the stimulus is transformed and then propagated as firing rate in the deeper layers.

Although the decoding accuracy and capacity were high in over half of the networks, the substantial decrease in the average values suggest that that information about the stimulus jitter can be encoded as rate but may be limited to four layers. However, because of experimental limitations, analyses were performed with a single neuron; accuracy and capacity are likely to improve substantially if many neurons are used for decoding.

**Transmission of rate information across layers**. To demonstrate further that rate information propagates, we took advantage of the high trial-to-trial variability in the firing rate evoked in the first layer (mean coefficient of variation ± SE = 0.71 ± 0.08, calculated by dividing the standard deviation of spike counts by the mean) and pooled the data obtained under all experimental conditions. For each network, we then sorted the evoked firing rates (calculated in a 300 ms time window after the light stimuli) in the first layer into five groups (each with an equal number of trials) and documented the associated rates in layers 2–4. The firing rate profiles (Fig. 5a) were separable in all layers, indicating that discriminability of rates is preserved (Fig. 5b). To compare across different experiments, the firing rates and PSTHs were normalized (see Methods; non-normalized data from a single experiment shown in Supplementary Fig. 12). Discriminability was also maintained when repetitive firing was induced in the neurons in the first layer by delivering long (0.5 s) Poisson trains of light pulses at different rates (Supplementary Fig. 4). Thus, information about firing rate propagated successfully.

To determine the range of frequencies that can be reliably propagated and discriminated, we constructed logistic maps where the abscissa is the firing rate in a given ($n$th) layer and the ordinate is the firing rate in the next ($n + 1$th) layer (Fig. 5c). For frequencies $\lesssim 15$ Hz, the curve (averaged from 16 networks) superimposed with the unitary slope line, indicating that in this range, firing rate was preserved across layers and uniquely represented. At higher frequencies, the curve became sublinear, which indicates that firing rate decreased to a fixed value of ~15 Hz within a few layers (dotted line in Fig. 5c).

To quantify discriminability in each layer, we trained a binary classification decision tree and defined the decoding accuracy as the probability of correctly identifying the firing rate in layer 1 based on the firing rate in the $n$th layer (see Methods). In successive layers, decoding accuracy for a single neuron decreased from a peak in layer 1 but remained well above chance in the deep layers (Fig. 5d). Similar results were found in simulated networks (Supplementary Fig. 13). To quantify how much information could be propagated in our system, we computed the channel capacity (Fig. 5e). This measure peaked at a value of about 2.1 bits in layer 1, which correspond to the maximum capacity of information that a single neuron is able to encode. In layers 2–4, channel capacity was constant and attained a value of about 0.7 bits (range: [0.1, 1.5] bits). Substantial improvement may be possible if decoding is performed with a population of cells: both

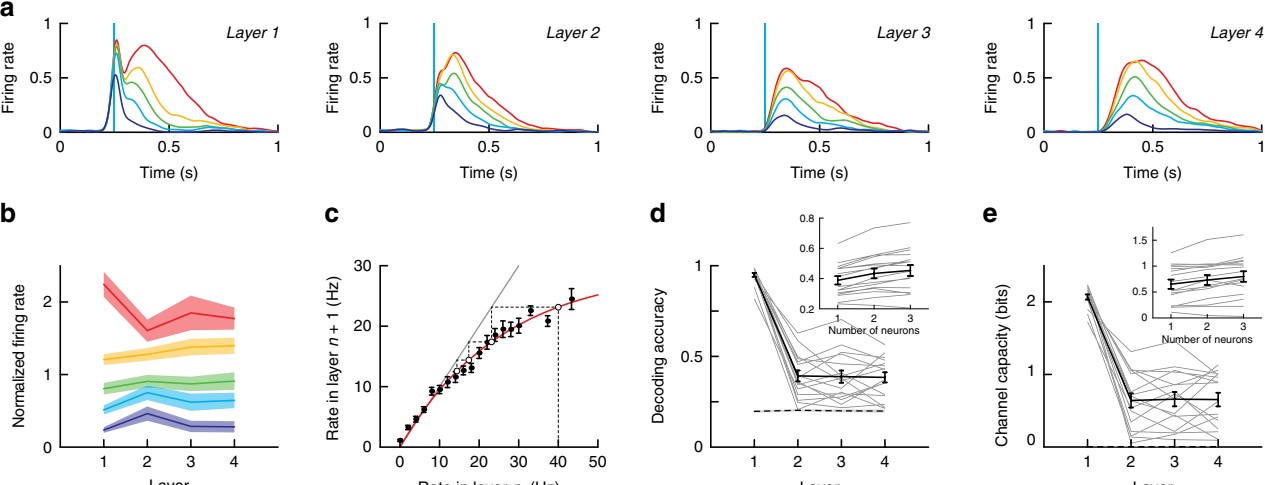

**Fig. 5** Propagation of firing rate information. **a** Average PSTHs in layers 1–4 (left to right) compiled from trials sorted in five groups based on firing rates in layers 1 (see text and Methods). **b** Firing rate vs layer. For each neuron of each network, the firing rate was normalized by the average rate across the different groups. **c** Logistic map showing output rate (=firing rate in layer $n + 1$) versus input rate (=firing rate in layer $n$). The gray line has a slope of 1 and the red curve is a fit to the data. Dotted lines represent a sample trajectory starting at 40 Hz in layer 1 and ending at 13 Hz in layer 4. **d** Decoding accuracy quantified as the probability of correctly classifying firing rate in the first layer given the firing rate in the $n$th layer. Inset shows the decoding accuracy when 1, 2 or 3 neurons from layers 2–4 where used simultaneously to decode the firing rate in layer 1. Paired $t$-test statistic for the increase in decoding accuracy with the number of neurons: 1 vs 2: $P = 1.1 \times 10^{-5}$, 1 vs 3: $P = 1.1 \times 10^{-4}$, 2 vs 3: $P = 0.012$. **e** Same as in **d** for the channel capacity. Paired $t$-test statistic for the increase in channel capacity with the number of neurons: 1 vs 2: $P = 0.002$, 1 vs 3: $P = 2.1 \times 10^{-4}$, 2 vs 3: $P = 0.004$. Data are compiled from the same $n = 16$ dense networks of Fig. 2 and are presented as mean ± SEM. In **d**, **e** all data are shown as thin lines

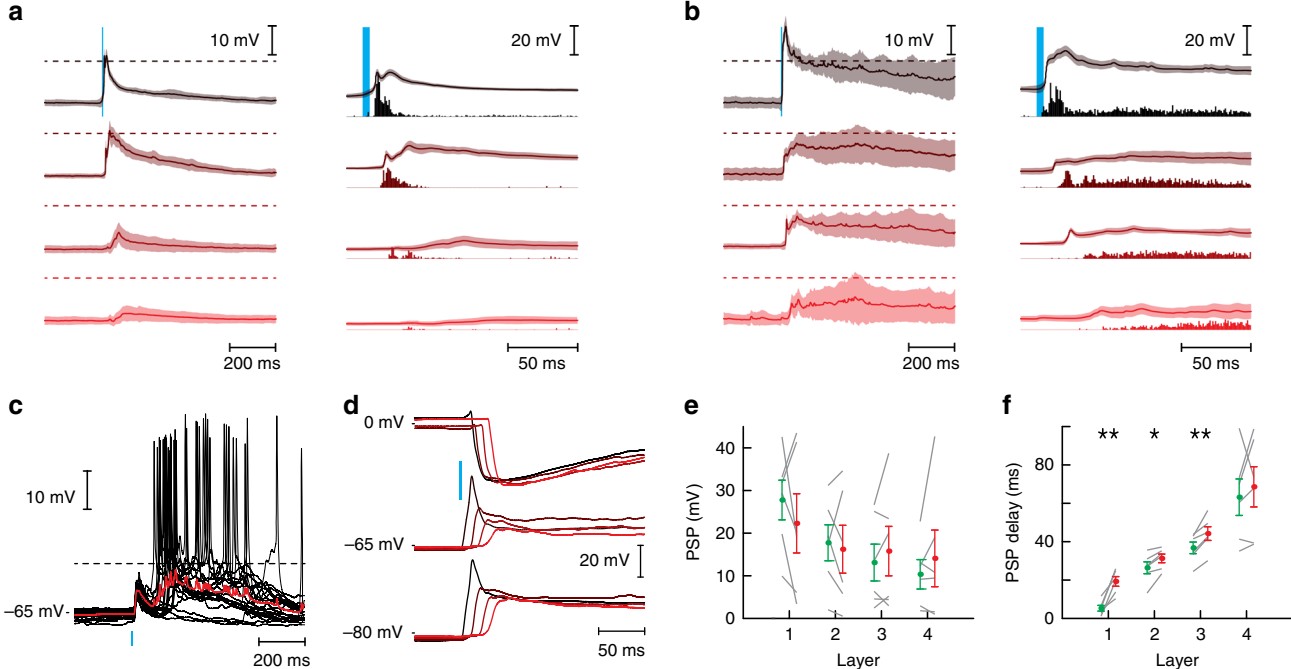

**Fig. 6** PSP dynamics in multilayer networks. **a** Average postsynaptic potential of neurons in sparse networks. Data were averaged from $n = 6$ networks (density $= 342 \pm 57$ neurons·mm$^{-2}$, mean $\pm$ STD) where a single neuron was recorded in layers 1–4 (black to red) of each network. The shaded area represents the standard deviation averaged over trials and then averaged over neurons. The dashed line represents the average spiking threshold of 15 mV above resting membrane potential[37]. Right: same data shown at higher temporal magnification and superimposed with the PSTHs from Fig. 2. **b** Same as in **a** but for dense networks ($n = 5$ networks, density $= 572 \pm 117$ neurons·mm$^{-2}$, mean $\pm$ STD). **c** Recordings from a neuron in layer 2 showing variability in the subthreshold potentials leading to threshold crossings (22 trials, mean membrane potential shown in red). **d** Example of excitatory (bottom), inhibitory (top), and compound (middle) post-synaptic potentials in a neuron when the light stimulus was delivered in layers 1–4 (black to red). **e** EPSP and IPSP amplitudes recorded simultaneously in layers 1–4 vs layer ($n = 6$ networks, density $= 510 \pm 131$ neurons·mm$^{-2}$, mean $\pm$ STD). Amplitudes between individual EPSPs and IPSPs were correlated (Pearson correlation: $R = 0.75$, $P = 2.1 \times 10^{-5}$, $n = 24$). Gray lines mark individual measurements. **f** PSP delay (stimulus onset to time at maximal slope) vs layer. Paired $t$-test statistic for the difference between E and I delays: L1: $P = 0.002$, L2: $P = 0.012$, L3: $P = 0.003$, L4: $P = 0.51$. $*P < 0.05$, $**P < 0.01$ In **e**, **f**, data are presented as mean $\pm$ SEM and individual data are shown as gray lines

the decoding accuracy (inset in Fig. 5d) and channel capacity (inset in Fig. 5e) improved when 3 neurons were used simultaneously for decoding.

**Maintaining E–I balance during propagation.** To examine the synaptic potentials underlying activity propagation, we performed whole-cell recordings from neurons in layer 1–4 during light stimulation in layer 1 (Fig. 6a, b). In the synfire mode, the postsynaptic potentials (PSPs) recorded in the first layer (Fig. 6a, black) rose sharply to a peak and decayed back to baseline. The PSPs evoked in the next layers had progressively longer onset times, slower rate of rise and fall, and smaller amplitudes. Superimposing the PSPs with the PSTHs shows that the action potentials were evoked on the rising edge of the PSPs (right).

In the rate mode, the underlying PSPs in the first layer rose rapidly to a peak and subsequently decayed at slow rate (Fig. 6b). There was a large decrease in PSP amplitude from the first to the second layer followed by much smaller decrease in subsequent layers. Compared to the synfire mode, the membrane potential hovered closer to firing threshold. Moreover, the membrane potential was more variable (shaded area is standard deviation across trials). The trial-to-trial variability allowed threshold crossings even when the average membrane potential was below threshold (Fig. 6c). The changes in the evoked firing rate across layer reflect the time course of the underlying PSP (Fig. 6b, right). In the first layer, the sharp rise in the PSP amplitude accounted for the transient firing while the slow decay produced the

persistent component. The transient spiking component disappeared with the fast component of the PSP, leaving only the slow component of the PSP and the persistent firing. In layers 3–4, the peak of the PSTHs lagged the peak of the PSP because a longer integration time was needed for the membrane potential to cross threshold.

The balanced regime—where the E and I synaptic inputs track each other both in magnitude[44] and in time[37,45,46]—is maintained during propagation within and across layers. To view the E and I synaptic potentials, we held the membrane potentials at $-80$ or 0 mV (reversal potentials of I and E, respectively) during repeated delivery of identical stimuli (Fig. 6d). Note that the isolation protocol for E reveals primarily the AMPA-mediated component, as the hyperpolarized potential likely did not release the Mg$^{2+}$ block of the NMDA-mediated receptors (see below). In the first layer, the large, sharp peak in the composite PSP measured at resting potential (Fig. 6d, middle, black) was due to a combination of a rapidly rising EPSP (bottom, black) and an IPSP (top, black). The IPSP increased the decay rate of the composite PSP but did not cancel the depolarization (Fig. 6d, middle). In subsequent layers, the EPSP and IPSP amplitudes both decreased (Fig. 6e) with the EPSPs decreasing at a slightly faster rate than the IPSPs. Nevertheless, consistent with the balanced regime, the IPSP increased nearly proportionally with the EPSP (Fig. 6e).

The relative timing of the AMPA-mediated EPSPs and IPSPs was also preserved across layers. We measured the delays of EPSPs and IPSPs at the maximal slope of the membrane potential. In the first layer, there was a relatively long delay

between the EPSPs and IPSPs, because only the E cells were stimulated and some time was needed for the inhibitory cells to fire (Fig. 6f). The result is that there is a time window where the edge of the EPSP can "escape" inhibition to evoke the early spikes that compose the transient phases of the synfire and rate mode. In layers 2–4, the EPSP-IPSP delay was significantly shorter as neurons could receive afferents from both E and I cells in the previous layer (Fig. 6f). With no sharp peaks in the composite PSP, the evoked action potentials were delayed (Fig. 2b, h). Thus, the amplitude, shape and arrival time of EPSPs were well matched by those of IPSPs, producing balance between excitation and inhibition that impeded further propagation of the transient component in deeper layers.

The E-I tracking and balance predicts that the noise correlation in spiking is low[45]. Low spiking correlation increases the effectiveness of rate codes[15]. To measure correlations, we performed cell-attached recordings from pairs of neurons within (Supplementary Fig. 14) and between layers. To measure noise correlation between neurons, we subtracted the 'signal' correlograms, constructed from shuffled trials, from the raw correlograms (see Supplementary Fig. 15). The noise correlation measured at the peak was low for neurons in the same layer (median value of $C = 0.03$; 1st and 3rd quartile $[-0.01, 0.17]$; $n = 30$ pairs; see Supplementary Fig. 16) and for neurons separated by 1, 2, or 3 layers (median values of $C = 0.01$; 1st and 3rd quartile $[-0.02, 0.08]$; $n = 96$ pairs).

**NMDA-mediated component of the prolonged activity.** The persistent phase in the PSP and firing was mediated in part by NMDA receptor synaptic current[47,48]. Using paired recordings, we confirmed the presence of strong NMDA-mediated components in the E synapses (Supplementary Fig. 17). The estimated ratio of NMDA to AMPA component in cultures (~0.7) was well within the range of those measured in vitro[49,50]. Blocking NMDA

currents with APV reduced substantially the duration of the evoked activity but had relatively little effect on the activity near the onset (<100 ms) (Fig. 7a, b left vs middle). This NMDA-insensitive component, which propagated to the 4th layer, was wider than the firing in the synfire mode and was likely mediated by recurrent activity. Adding GABA blocker (bicuculline) increased but did not fully reproduce the original firing activity: the duration of the responses recovered (Fig. 7c) but the total activity was larger than in the control conditions (Fig. 7d), indicating that recurrent activity was participating but was insufficient to produce the full extent of the activity.

The NMDA component, by prolonging the activity, enhanced the transmission of information. In the presence of APV, the channel capacity in layers 2–4 decreased by half ($0.39 \pm 0.07$ bits, mean ± SE in neurons from layers 3 to 4) as compared to control ($0.80 \pm 0.11$ bits). The decreased channel capacity was unlikely due to the decrease in overall firing rate (Fig. 7d) as the channel capacity in layer 1 were similar both in the absence and presence of APV (Fig. 7e). By the same token, blocking inhibition, which raised the firing rate (Fig. 7d), increased channel capacity ($0.64 \pm 0.09$ bits) but did not reach control levels. These results suggest that both the late response carried by the NMDA current and a larger gain provided by recurrent excitation play a role in the propagation of firing rate information.

These results were qualitatively reproduced by simulations. In simulated networks with only feedforward connections, brief pulse packets propagated (Fig. 8a) provided there was a sufficient number of neurons per layer, as predicted by theory[10]. Adding recurrent and feedback excitatory connections respectively within and across layers evoked activity with an initial transient phase followed by a sustained phase (Fig. 8b). The addition of inhibitory neurons prevented propagation even with the recurrent connectivity (Fig. 8c) because inhibitory connections were both strong and dense. Propagation was rescued when NMDA-

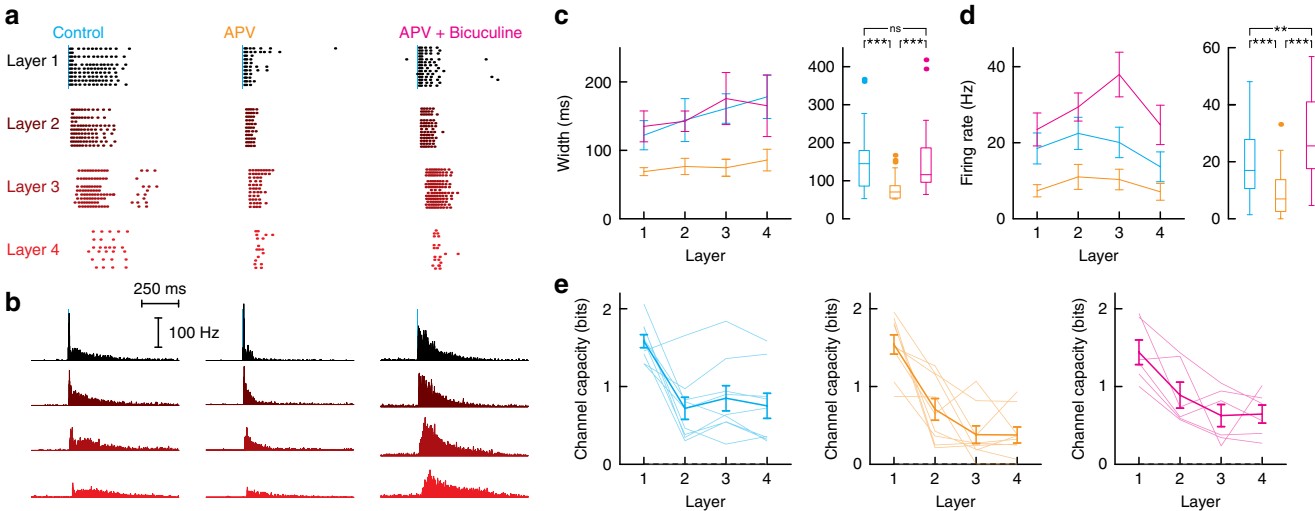

**Fig. 7** Roles of AMPA and NMDA-mediated components in the propagation of activity. **a** Example of dot rasters of spikes recorded from individual neurons in each layer (layers 1–4 from black to red) in cell-attached mode in response to synchronized activation of neurons in the first layer (30 repetitions) under control conditions (left), in the presence of 50 μM APV, and in the presence of APV and 10 μM bicuculline (right). **b** Average poststimulus time histogram in each layer. Data were compiled from 9 networks of density = 278 ± 68 neurons·mm⁻² (mean ± STD). **c** Left, average firing rate half width vs layer (mean ± SEM). Right, data combined across layers shown as whisker plots (box: median and interquartile range, whiskers: full range of the distribution; outliers are plotted individually). **d** Similar plots for the firing rate. Mann–Whitney U-test statistics for the width: control vs APV: $P = 7.5 \times 10^{-6}$, APV vs Bicuculine: $P = 2.7 \times 10^{-6}$, control vs Bicuculine: $P = 0.85$. Mann–Whitney U-test statistics for the firing rate: control vs APV: $P = 3.2 \times 10^{-4}$, APV vs Bicuculine: $P = 1.1 \times 10^{-7}$, control vs Bicuculine: $P = 0.008$. ns not significant, $*P < 0.05$, $**P < 0.01$, $***P < 0.001$. **e** Channel capacity quantified as the probability of correctly classifying firing rate in the first layer given the firing rate in the nth layer. All data are shown and the mean ± SEM is represented as thick lines. Mann–Whitney U-test statistic for the comparison of decoding accuracy of neurons in layers 2–4 vs conditions: control vs APV: $P = 0.010$, APV vs Bicuculine: $P = 0.032$, control vs Bicuculine: $P = 0.954$.

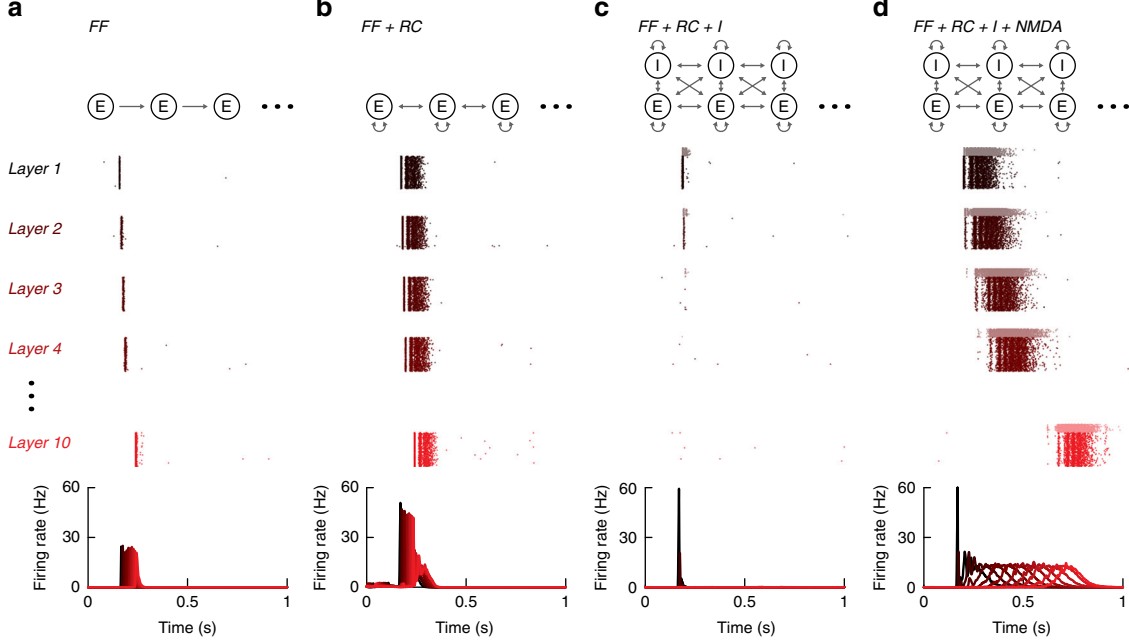

**Fig. 8** Simulation of multilayer networks. **a** Dot rasters of spikes showing simulations of a purely feedforward network composed of only excitatory neurons (480 neurons; see Methods for a detailed description of the network and parameters). All neurons of a single trial are shown in the dot raster (top) and 30 trials are compiled to plot the average firing rate (bottom). **b** Same network as in **a** but with additional recurrent connections between and within layers. **c** Same network as in **b** with inhibitory neurons (120 inhibitory neurons, i.e. 20%, presented as light dots in the spike raster). **d** Same network as in **c** but with NMDA-mediated synaptic current

mediated synaptic current was included (Fig. 8d). Consistent with the rate mode in the experiments, there was a prominent persistent component which did not show a substantial increase in synchrony across layers beyond what was already present in layer 1 (compare with[4,9,13]). Some synchrony was present in layer 1 probably because the model did not capture the variability observed in the cultures (Fig. 6c, Supplementary Fig. 14).

**Propagation in layers of different sizes**. The fact that propagation occurs more readily in dense networks, which contains more recurrent connections, than in sparse networks suggests that the direction of propagation may be biased in the direction of increasing network size. In cortex, for example, the number of neurons and the number of connections in progressively higher order regions vary significantly[51,52].

To examine direction bias, we constructed a multilayer culture network where the area of each compartment increases in one direction (Fig. 9a). Using the spatial profile of connection probability (Supplementary Fig. 2), we estimated that the number of connections $K$ increases almost linearly from the 1st to the 4th layer by three-fold ($K = 50$ to $K = 140$ in layer 1 and 4, respectively, Supplementary Fig. 1). Given the synaptic scaling $J \propto 1/\sqrt{K}$ (where $J$ is the synaptic strength) that occurs in cortical neurons in culture[37], the total input $\mu = J \times K \times r$ (where $r$ is the average firing rate) is expected to scale as $\sqrt{K}$ and thus to increase in large networks.

To test for direction bias, neurons in the small layer were stimulated synchronously and the activity of neurons monitored in the direction of increasing layer size (Fig. 9b). Then, an equal number of neurons were stimulated in the large end and spiking activity monitored in the direction of decreasing layer size (Fig. 9c). Because the rate mode propagated reliably, we focused here on the synfire mode. As predicted, propagation occurred more readily in the direction of increasing layer size, as evidenced by

the increased spike probability and firing rate in the last layer (Fig. 9d, e). The delays of action potentials were slightly shorter in the direction of increasing size (Fig. 9f), consistent with the fact that small networks resembled sparse networks. Taken together, these results suggest that the direction of signal transmission could be biased in non-homogeneous network by amplification of signal in the direction of increasing size.

## Discussion

We examined signal propagation using a multi-layered culture preparation consisting of cortical neurons. We found that contrary to predictions of theory[8,10,11] and experiments with iteratively-constructed networks[9], the propagated activity did not evolve to a fully synchronous state. Rather, the evoked firing actually became more temporally dispersed across layers. This feature allowed rate signals to propagate successfully with low correlations without the need to introduce large background noise[13,16]. Rate propagation occurred because the strong NMDA component of the excitatory input prolonged the decay of the synaptic potential and kept the membrane potential near threshold, resulting in highly variable spiking activity across trials (Figs. 2b, 4a, 6b, c, 7a) and across simultaneously recorded cells (Supplementary Fig. 14). The net effect is to maintain decoding accuracy and channel capacity through the layers (Fig. 5d, e and Fig. 7e).

These results complement and extend previous findings obtained in culture preparations using extracellular recording arrays or calcium imaging[33,34,36]. Indeed, rate information contained in spontaneous or stimulus-evoked bursts propagated through a network of neurons cultured in a long, tubular chamber over distances of 3 mm, or ~10 axonal lengths[36]. The high temporal and spatial resolution afforded by using multilayer cultured networks in combination with optogenetic stimulation and whole-cell recording allowed us to examine the underlying synaptic and network mechanisms. Importantly, the presence of

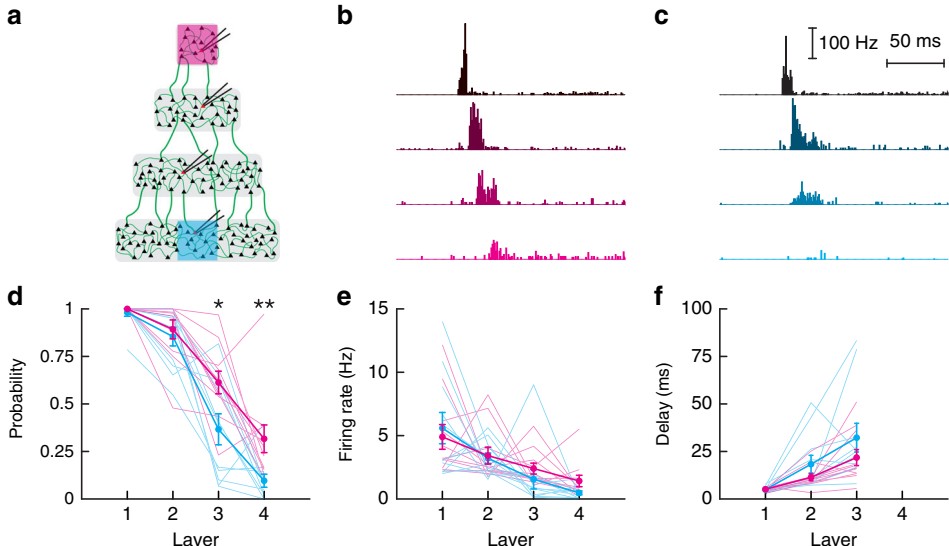

**Fig. 9** Propagation in networks of different sizes. **a** Schematic of the multilayer network with compartments of different sizes. All layers were 0.5 mm in width with lengths of 0.5, 1, 1.5, and 2 mm from top to bottom. An area of same dimension was photostimulated either in the smallest (magenta) or in the largest (cyan) compartment. **b**, **c** Average poststimulus time histogram of neurons when the small (**b**) or the large (**c**) network was stimulated. In each network, a single neuron is recorded in each layer (from black to magenta (cyan) for layers 1–4, respectively). Data were averaged from $n = 11$ networks of density $= 297 \pm 111$ neurons·mm$^{-2}$ (mean ± STD). **d**–**f** Spike probability (**d**), firing rate (**e**), and delay of first spike (**f**) as a function of layer when the smallest (magenta) or the largest (blue) network was activated. In **d**–**f** all data are shown and the mean ± SEM is represented as thick lines. Paired Mann–Whitney $U$-test statistics: **d** L1: $P = 1$, L2: $P = 0.643$, L3: $P = 0.024$, L4: $P = 0.019$; **e** L1: $P = 0.898$, L2: $P = 0.898$, L3: $P = 0.102$, L4: $P = 0.067$; **f** L1: $P = 0.966$, L2: $P = 0.465$, L3: $P = 0.278$

distinct layers facilitated direct testing of seminal theories of signal propagation.

The results deviated from theoretical predictions largely because of the contribution of the NMDA component but also because of differences in network architecture. Unlike idealized feedforward networks used in most models, the multilayer culture network contained feedback and recurrent connections. However, these interlayer connections are unlikely to contribute significantly to rate propagation. When NMDA was blocked with APV, the response rate and duration decreased significantly, as did the information capacity (Fig. 7). The remaining response fraction was mostly due to recurrent connections within a layer rather than across layers: based on the connection profile, a neuron receives 80% of its connections from the same layer and only 20% from the neighboring layers (i.e. 10% from feedforward inputs from previous layer and 10% from feedback inputs from the next layer). We used simulations to further understand the impact of the different connection types (Supplementary Fig. 18). In purely feedforward networks made of excitatory and inhibitory neurons; propagation failed after few layers. Whereas the addition of feedback connections to the network did not qualitatively change the behavior, the presence of recurrent connections strongly supported propagation.

In the framework of information theory, the channel capacity of a neuron is related to the range of firing that the neuron can explore. Because noise interferes with the discrimination of nearby levels of firing activity, the capability to generate a broad range of firing is important for representing information as rate: networks that have a narrow range of responses to a wide range of stimuli, for example, have limited information capacity. Previous studies have shown that spontaneous[53,54] and evoked[36] firing in cultures are not all-or-none but rather exhibit a range of firing rates. We showed that the firing rate is modulated by subtle changes in the rate, number, and timing of the light pulses delivered to the network. Small jitter generated large firing rates probably because the large depolarization caused by the near

synchronous arrival of synaptic potentials increased the NMDA component, which in turn kept the membrane potential closer to threshold for a longer period of time and recruited more cells in the network. The transformation of temporal code to a rate code and subsequent propagation through layers is of some functional significance: for example, phase-locked, temporally precise signals generated by specialized brainstem neurons are lost en route to cortex[5–7,25,28].

The culture preparation is versatile and has been used previously to engineer complex networks[35]. However, a major concern with the reduced preparation is whether the networks can reproduce firing behavior observed in the intact animals. Though the proportion and patterns of connections between excitatory to inhibitory neurons is similar to those in intact brains[55–57], the detailed microcircuitry is lost. Yet, despite these differences, the firing behavior of cultured networks with a wide range of densities reproduces salient properties of neurons in vivo[37], including low spiking correlation[58] and stimulus-induced decreases in variability[59]. We stress that these results are general and are not restricted to the culture preparation: the results were obtained with no fine tuning of the culture or stimulus parameters and indeed were reproducible with a wide range of conditions. The NMDA-dependent component that occurs in the multilayer cultured networks is similar to the late response observed in sensory systems which is also sensitive to perturbation of NMDA current[31,32] and may involve regenerative processes (so-called NMDA spikes[60,61]). Moreover, the increased duration of firing resembles prolonged firing observed in sequentially higher order brain structures (visual: ref. [22]; auditory: refs. [23,24]; somatosensory: refs. [5,25]; frontal and temporal cortex: refs. [26,27]). In rodents, a brief deflection of the whisker results in a transient response in the thalamorecipient layer 4 and in a more sustained response in layer 2/3, with a substantial reduction in the transient component[5,25,29].

Propagation of rate in the cultured networks depended on several conditions that are likely to be satisfied in the intact brain.

First, the network has to be sufficiently dense to support recurrent activity. The density that supported rate propagation was ~500 neurons·mm$^{-2}$ (Fig. 2f) indicating an average of 400 connections per neuron[37], much less than that estimated for pyramidal cells in vivo[55,62]. Second, the NMDA-mediated component of the excitatory synapse must be sufficiently large to maintain the membrane potential near threshold. The time course and magnitude of the NMDA-mediated component in the cultures are comparable to those measured in vitro[49,50]. Finally, the ratio of inhibitory to excitatory neurons (0.23; ref. [37]) is similar to those estimated in vitro and in vivo; the inhibitory amplitude is larger in the cultures but scales with network size and in proportion to AMPA-mediated components[37]. Although inhibition did not track completely the much longer-lasting NMDA component, the network did not result in runaway excitation, indicating that overall excitatory–inhibitory balance was maintained.

Feedforward networks are a primary means of communication between regions of the nervous system. However, theoretical analyses predict that the signals that propagate default to synchronous events. With the NMDA-mediated propagation, both rate and temporal (transformed as rate) information can be transmitted at least across four layers. An intriguing consequence is that the signals that are propagated could be potentially controlled by neuromodulation of NMDA[63].

## Methods

**Primary neuron cultures.** Dissociated cortical neurons from postnatal (P0–P1) mice of either sex were prepared as described previously[37,64] and in accordance with guidelines of the New York University Animal Welfare Committee. Briefly, the mouse cortex was dissected in cold CMF-HBSS ($Ca^{2+}$ and $Mg^{2+}$ free Hank's balanced salt solution containing 1 mM pyruvate,15 mM HEPES, 10 mM $NaHCO_3$). The tissue was dissociated in papain (15 U·mL$^{-1}$, Roche) containing 1 mM L-cystein, 5 mM 2-amino-5-phosphonopentanoic acid and 100 U·mL$^{-1}$ DNase (DN25; Sigma) for 25 min. After enzymatic inactivation in CMF-HBSS containing 100 mg·mL$^{-1}$ BSA (A9418; Sigma) and 40 mg·mL$^{-1}$ trypsin inhibitor (T9253; Sigma), pieces were mechanically dissociated with a pipette. Cell concentration was measured before plating using a haemocytometer. Approximately $0.3–3 \times 10^6$ cells were plated on each coverslip, resulting in a density of ~100–1000 cells·mm$^{-2}$ at the time of experiment. Neurons were seeded onto German glass coverslips (25 mm, #1 thickness, Electron Microscopy Science). Glass was cleaned in 3 N HCl for 48 h and immersed in sterile aqueous solution of 0.1 mg·mL$^{-1}$ poly-L-lysine (MW: 70,000–150,000; Sigma) in 0.1 M borate buffer for 12 h. Neurons were grown in Neurobasal medium (supplemented with B27, Glutamax and penicillin/streptomycin cocktail; Invitrogen) in a humidified incubator at 37 °C, 5% $CO_2$. One third of the culture medium was exchanged every 3 days.

Expression of channelrhodopsin (ChR2) in excitatory neurons was achieved by crossing homozygote *Vglut2-Cre* mice (016963, Jackson Laboratory) with *ChR2-loxP* mice (Ai32, 012569, Jackson Laboratory). Experiments were performed at 14–21 DIV, when neuronal characteristics and network connectivity were stable and expression of ChR2 was sufficient to enable reliable photostimulation.

**Microfabrication and microchambers' production.** The culture chambers had rectangular compartments (width $w$, length $l$) separated by a gap of length $c$. Neurons were confined to the layers and axons grew bidirectionally through the gaps. We designed a symmetric network where all layers had the same size (Fig. 1b; in mm: $w = 0.7$; $l = 6$; $c = 0.4$) and an asymmetric network where the length increased linearly with layer number (Fig. 8a; in mm: $w = 0.5$; $l = 0.5, 1, 1.5, 2, c = 0.4$) (see also Supplementary Fig. 1b). The symmetric network was made of six layers and we recorded neurons in layers 2–5 (and not in layer 1 or 6) to avoid edge effects due to fewer connections at the terminal layers (see Supplementary Fig. 1). The asymmetric network had seven layers and we recorded from layer 1 to 4 to maximize the variation of number of connections (see Supplementary Fig. 1b).

The chambers were made in PDMS using soft lithography and replica molding. To fabricate the master with positive relief patterns of cell culture compartments, we built a single layer of photoresist of 160 μm in height. A layer of SU82050 was spin-coated onto the wafer at 1200 rpm and then soft-baked for 7 min at 65 °C and 30 min at 95 °C. The template was then exposed to UV light through an optic plastic mask (CAD/Art Services) of the culture compartment. After hard bake (5 min at 65 °C, 12 min at 95 °C, and 1 min at 65 °C), the final mold was developed in SU8 developer.

We casted and cured a polydimethylsiloxane polymer (PDMS, Sylgard 184, Dow Corning) against the positive relief master to obtain a negative replica-molded piece. PDMS was mixed with curing agent (10:1 ratio) and degassed under vacuum. The resulting preparation was poured onto the mold, pressed between two glass

slides and cured at 110 °C for 2 min onto a hot plate. After curing, the PDMS piece was peeled away, sterilized with ethanol and sealed onto the treated glass coverslip. The resulting assembly was washed with PBS and incubated at 37 °C overnight. After rinsing, the device was flooded by culture medium. Neurons were added and cultured normally. After 24 h, the PDMS mold was peeled away from the glass coverslip to allow processes to grow and connect different layers.

**Electrophysiological recordings.** Recordings were performed at room temperature in HEPES-based artificial cerebrospinal fluid (aCSF). The aCSF solution contained (in mM): 125 NaCl, 10 $NaHCO_3$, 25 D-glucose, 2.5 KCl, 2 $CaCl_2$, 1.25 $NaH_2PO_4$, 1 $MgCl_2$, and 10 HEPES. For some experiments, 50 μM APV was added to block the NMDA component of postsynaptic currents or 10 μM bicuculline to block GABA-A inhibitory currents.

Electrodes, pulled from borosilicate pipettes (1.5 OD) on a Flaming/Brown micropipette puller (Sutter Instruments), had resistances in the range of 6–10 MΩ when filled with internal solution containing (in mM): 130 κ-gluconate, 10 HEPES, 10 phosphocreatine, 5 KCl, 1 $MgCl_2$, 4 ATP-Mg, and 0.3 mM GTP.

Cells were visualized through a × 10 water-immersion objective using infrared differential interference contrast (IR-DIC) and fluorescence microscopy (BX51, Olympus). Simultaneous whole-cell current-clamp recordings were made from up to four neurons using BVC-700A amplifiers (Dagan). The signal was filtered at 5 kHz and digitized at 25 kHz using an 18-bits interface card (PCI-6289, National Instrument). Signal generation and acquisition were here and in the following controlled by a custom user interface programmed with LabVIEW (National Instrument).

**Optical stimulation setup.** We used a Digital Light Processing projector (DLP LightCrafter; Texas Instrument) to stimulate optically neurons expressing ChR2 as previously described[37,41]. The projector had a resolution of 608 × 684 pixels. The image of the projector was de-magnified and collimated using a pair of achromatic doublet lenses (35 mm and 200 mm; Thorlabs). A dual port intermediate unit (U-DP, Olympus) was placed in-between the fluorescent port and the projection lens of the microscope and enclosed a 510 nm dichroic mirror (T510LPXRXT, Chroma). The resulting pixel size at the sample plane was a rectangle of dimensions 2.2 μm × 1.1 μm. We used the inbuilt blue LED of the projector which has a center wavelength of 460 nm and intensity of 10 mW·mm$^{-2}$ at the sample plane. The time resolution of the projector was 1440 Hz.

**Stimulation and recordings protocols.** We first selected $N_{stim} = 20–40$ regions of interest (ROIs) that were drawn onto ChR2 positive neurons (about 10–20% of neurons that are in the field of stimulation). Care was taken to stimulate small areas of ~30 μm × 30 μm to avoid stimulation of processes belonging to adjacent neurons. Each ROI was stimulated by a single light pulse of duration $\Delta_{pulse} = 5$ ms. The light intensity was fixed at 10 mW·mm$^{-2}$; a value that was sufficient to evoke reliably a spike in the selected neurons but too low to stimulate neighboring cells. Particular attention was paid to record neurons that do not express any ChR2 to avoid any obvious cross-activation.

In most experiments we varied the temporal jitter in the light pulses delivered to individual neurons. The jittered pulses (termed pulse packet) were Gaussian distributed and the standard deviation was varied ($\sigma = 0, 5, 10, 15, 20$ ms). In some experiments we delivered 500 ms-long Poisson trains of light pulses to the first layer and we varied the effective pulse rates (5, 10, and 20 Hz; Supplementary Fig. 4). A given stimulus was repeated $N_{trials} = 5–6$ times for PSPs data and $N_{trials} = 10–40$ times for spike data. We allowed at least 5 s of recovery between each stimulation. In each network, we performed simultaneous cell-attached or whole-cell recordings from four neurons (one neuron in each layer) in current clamp mode.

**Data analysis.** Analysis of network characteristics: From IR-DIC images of the recording site taken after every experiment, neuron density $d$ was estimated by counting somata on a ~1 × 1 mm$^2$ area in each layer. Data were pooled according to densities. Low and high densities corresponded to networks of neuronal densities (in neurons·mm$^{-2}$): $d < 450$, and $450 < d$, respectively.

Analysis of membrane potential data: The evoked postsynaptic potentials (PSPs) were averaged over 5–6 stimulus repetitions. The amplitude was measured at the peak value (or trough for inhibitory PSPs) and the PSP delay at the time of maximal slope (Fig. 6). We used the Pearson correlation coefficient to quantify the correlation between IPSPs and EPSPs (Fig. 6e). The significance of the correlation coefficient was determined using Student's $t$-distribution.

Analysis of spike data: The spike probability was defined as the probability of observing at least one spike on a given trial (Fig. 2e, f). The delay was estimated from the time of the first spike at each trial (Fig. 2g). The spike spread was measured as the standard deviation of first spike times across trial (Fig. 2h). The firing rate was defined as the average firing rate in a window of 300 ms following the stimulus, as mentioned in the main text (Figs. 2e, f, 4b, 5b, and 7c). Firing rates for poststimulus time histogram were computed by convolving the spikes data with a Gaussian kernel of width 25 ms (Figs. 4a and 5a). PSTHs represented as bar plots had a binning window of 1 ms (Figs. 2c,d, 6a, b, and 7b).

| **Table 1 Intrinsic parameters of leaky integrate-and-fire neuronal models** | | | | | | |
|---|---|---|---|---|---|---|
| Neuron type | $R_m$ (MΩ) | $\tau_m$ (ms) | $V_{leak}$ (mV) | $V_t$ (mV) | $V_{reset}$ (mV) | $\tau_{ref}$ (ms) |
| Excitatory | 190 ± 90 | 26 ± 16 | −60 | −44 ± 6 | $V_t$ − 12 | 30 ± 6 |
| Inhibitory | 135 ± 70 | 17 ± 11 | −60 | −46 ± 5 | $V_t$ − 12 | 20 ± 4 |

| **Table 2 Connectivity in the simulated multilayer network** | | | | | |
|---|---|---|---|---|---|
| Post-synaptic<br>Presynaptic | Excitatory | | | Inhibitory | |
| Excitatory | $g_{AMPA}$ = 10 nS<br>$g_{NMDA}$ = 30 nS | $P_{within}$ = 0.29<br>$P_{between}$ = 0.07 | | $g_{AMPA}$ = 12 nS<br>$g_{NMDA}$ = 36 nS | $P_{within}$ = 0.4<br>$P_{between}$ = 0.11 |
| Inhibitory | $g_{GABA}$ = 36 nS | $P_{within}$ = 0.3<br>$P_{between}$ = 0.03 | | $g_{GABA}$ = 46 nS | $P_{within}$ = 0.25<br>$P_{between}$ = 0.02 |

To examine propagation of rate signals (Fig. 5a, b) across layers, the firing rates evoked in the first layer of each network were divided into five groups (highest rate was group 1, lowest group 5). Because the range of firing rates evoked in each network could vary substantially (Fig. 2e), the data obtained from individual network were normalized before pooling and averaging. The PSTHs of each network at all layers were divided by the maximum firing rate in group 1 of the corresponding layer prior to averaging (Fig. 5a). To display the average firing rate compiled from the pooled data across layers (Fig. 5b), the evoked firing rate of each network was divided by the average of the firing rates in the 5 groups. With these normalization procedures, the separation of the five firing rate groups across layers could be readily visualized.

Decoding and mutual information: To decode temporal information (i.e. the correct stimulus among the different jitters $\sigma_j \in \Sigma = [0, 5, 10, 15, 20$ ms]), we trained a binary classification decision tree for each neuron of each network to identify the stimulus using two parameters: the firing rate in the 300 ms following the stimulus and the time of the first spike (Fig. 4c). This allowed us to estimate the conditional probability $p\left(r_i|\sigma_j\right)$ of attributing the response $r_i$ to stimulus $\sigma_i$ given the stimulus $\sigma_j$. We used the Statistics and Machine Learning Toolbox from Matlab to train the classifier. To avoid overfitting, we limited the number of leaves in the tree to the number of classes to decode (5). We also cross-validated our decoder: the classifier was trained using 80% of the data and tested using the remaining 20%. This procedure was repeated 50 times on randomly selected sets of data to estimate the average confusion matrix $\boldsymbol{M} = \left[p\left(r_i|\sigma_j\right)\right]_{i,j}$. We defined the decoding accuracy $d$ as the proportion of well-attributed trials by the decoder: $d = trace(\boldsymbol{M})$. We used $t$-tests to assess the statistical difference between the distribution of these 50 estimated decoding accuracies and the 50 ones measured when trials were randomized. This allowed us to determine how many neurons had a statistically significant decoding accuracy with a $p$-value of 0.01 (Fig. 4c and Supplementary Fig. 10c).

We used the non-uniform partitioning obtained from the classification results as the base to compute mutual information. Our method is similar to adaptive partitioning of the $(R, \Sigma)$ space[65]. By maximizing decoding accuracy and thus the sum of diagonal elements of the confusion matrix $p(r_l|\sigma)$, our partitioning reduces the number of empty entries and limits bias in the estimation of mutual information. The mutual information $MI_l$ between the stimulus and the firing rate in layer $l$ was calculated as follows:

$$MI = \sum_{i,j} p\left(r_i|\sigma_j\right) \cdot \log_2\left(\frac{p\left(r_i|\sigma_j\right)}{p\left(r_i\right) \cdot p\left(\sigma_i\right)}\right) \quad (1)$$

where $p(\sigma_i)$ and $p(r_i)$ are the probability distributions of responses and stimuli, respectively, and $p\left(r_i|\sigma_j\right)$ is the stimulus conditional probability distributions of responses. In principle, the upper bound for mutual information is the entropy of the stimulus which equals $\log_2(5) = 2.32$ bits because the stimulus can take 5 different values. The channel capacity was then computed numerically using the Arimoto-Blahut algorithm[66]. This method maximizes the mutual information and provides an estimate of how much information can be propagated between the applied stimulus and the observed response.

In a second analysis, we used the firing rate as the sole feature for the decoder and calculated mutual information and channel capacity in the same way (Fig. 4d).

To estimate how much information of the firing rate was propagated (Fig. 5), we ordered and grouped the firing rates measured in layer 1 into five non-overlapping groups and defined each group as a stimulus $s$. This was done for each network separately. We used the number of groups (5) to be the same as the

precedent analysis to compare quantitatively both results. Similarly, we the trained a classifier to discriminate and classify the stimuli $s$ according to the firing rates in the layers 1–4. We measured the decoding accuracy and the channel capacity as described above.

**Simulation of multilayer networks**. The simulated multilayer network consists of five layers, with $N \in [80, 960]$ neurons in each layer. The proportion of inhibitory neurons was set to 20%. We used leaky integrate-and-fire neuron model. The membrane potential $V(t)$ of each neuron was governed by the following equation:

$$\tau_m \dot{V}(t) = -V(t) + R_m(I_{noise}(t) + I_{AMPA}(t) + I_{NMDA}(t) + I_{GABA}(t)) \quad (2)$$

where $R_m$ is the input resistance, $\tau_m$ is the membrane time constants, $I_{noise}(t)$ is a noisy input current, and $I_{AMPA}(t)$, $I_{NMDA}(t)$, and $I_{GABA}(t)$ are synaptic currents. A spike was generated when $V$ exceeded the voltage threshold $V_t$ and was then reset to $V_{reset}$. No other spike could occur during the refractory period $\tau_{ref}$. Parameters and distributions were derived from experimental measurements[37] and are reported in Table 1. Equations were integrated using the Euler method and a step time dt = 50 μs.

Neuronal units were subjected to fluctuations in their input current which obeys the following equation:

$$I_{noise}(t) = \sqrt{C} \cdot I_{com}(t) + \sqrt{1 - C} \cdot I_{ind}(t) \quad (3)$$

where $I_{com}(t)$ is a noisy input common to all simulated neurons, $I_{ind}(t)$ is an independent noisy input to each neuron, and $C = 0.1$ is a constant.

$I_{com}(t)$ and $I_{ind}(t)$ were realizations of a Gaussian (Ornstein-Uhlenbeck) noise process and were generated using:

$$I(t_{n+1}) = I(t_n) \cdot e^{-dt/\tau} + \sigma \cdot \sqrt{1 - e^{-2dt/\tau}} \cdot \xi(t_n) \quad (4)$$

where $I(T_0) = 0$ is the initial condition, $\sigma = 15$ pA is the standard deviation, $\tau$ is the correlation time ($\tau = 15$ ms for the independent component and $\tau = 30$ ms for the common component) and $\xi(t_n)$ is a random variable drawn from the standard normal distribution of zero mean and unity variance.

The neuronal architecture was based on connection probabilities between and within layers that was derived from experimental measurements (Supplementary Fig. 2 and ref. [37]). To simplify the architecture and compare with existing data, we simulated a multilayer network with a width of about 1.5 mm in which we considered that the connection probability was constant (Table 2). With these settings, the simulated area corresponds to $1.5 \times 0.7 \approx 1$ mm$^2$, such that the number of neurons per layer in simulations is directly comparable to the neuronal density in experiments.

We used conductance-based synapses. Each synapse is described by a conductance variable $g$ that obeys the following equation:

$$g(t) = g_{syn}\frac{\tau_d\tau_r}{\tau_d - \tau_r}\left(\exp\left(-\frac{t - t_s - t_d}{\tau_d}\right) - \exp\left(-\frac{t - t_s - t_d}{\tau_r}\right)\right) \quad (5)$$

where $g_{syn}$ is the synaptic conductance (AMPA, NMDA, GABA), $t_s$ is the time of spike, $t_d$ is the synaptic delay, and $\tau_r$ and $\tau_d$ are the rise and decay time constants, respectively. The synaptic delay is 3 ms within layer and 8 ms between layers. The rise and decay times are the same for AMPA and GABA-based currents and were $\tau_r = 0.1$ ms and $\tau_d = 15$ ms. For NMDA-based currents, rise and decay time constants were $\tau_r = 5$ ms and $\tau_d = 100$ ms. The synaptic conductances given in Table 1 yield amplitudes of about 2.3 mV and −2.5 mV for EPSPs and IPSPs, respectively. Synaptic conductances were scaled according to the scaling rule that we measured in the neuronal culture preparation: $g = g_0/\sqrt{N/320}$[37].

Thus, the total synaptic current that neuron $i$ receives is:

$$I_{\text{AMPA},i}(t) = \sum_j s_j(t) \cdot p_{i,j} \cdot g_{i,j}(t) \cdot (V_i(t) - V_E) \tag{6}$$

$$I_{\text{NMDA},i}(t) = \sum_j s_j(t) \cdot p_{i,j} \cdot g_{i,j}(t) \cdot (V_i(t) - V_E) / \left(1 + \frac{\exp(-0.062 \cdot V_i(t))}{3.57}\right) \tag{7}$$

$$I_{\text{GABA},i}(t) = \sum_j s_j(t) \cdot p_{i,j} \cdot g_{i,j}(t) \cdot (V_i(t) - V_I) \tag{8}$$

where $s(t)$ is a synaptic depression variable, $p_{i,j} = \{0, 1\}$ is a binary variable of the connectivity matrix, $V_i(t)$ is the membrane potential. The equation for NMDA-based currents is including the voltage dependency of NMDA receptors.

Synaptic transmission was subjected to depression as follow. The excitatory (inhibitory) synaptic conductance was set to 35 % (30 %) of its previous value after a spike and recovered to its initial value according to:

$$\tau_s \dot{s}(t) = 1 - s(t) \tag{9}$$

where $\tau_s$ is the recovery time constant and was 850 ms for excitatory and 400 ms for inhibitory synapses based on recording in the culture preparation (Supplementary Fig. 19).

To stimulate the network, we provided a 1 nA current pulse during 5 ms in 30% of excitatory neurons randomly selected from the 1st layer. We varied the jitter in the same way as in experiments. To analyzed data, we randomly selected 20 neurons in each layer and performed the same analysis as for experiments.

**Statistical analysis**. The number of neurons and the number of experiments (i.e. the number of networks) that were used in each figure are shown in the Supplementary Table 1. All the data were shown as mean ± SEM., unless stated otherwise. No assumptions of normality of data distributions were imposed. Two group comparisons were performed using either paired or unpaired two-sided Mann–Whitney $U$-test. $t$-tests were used to compare decoding accuracies with the shuffle estimate (Fig. 4c), increase in decoding accuracy and channel capacity upon pooling (insets in Fig. 5d, e), and the delay between EPSP and IPSP (Fig. 6f). The variances between groups were assumed to be different. No statistical methods were used to pre-determine sample sizes.

**Reporting summary**. Further information on research design is available in the Nature Research Reporting Summary linked to this article.

## Data availability
The datasets generated and analyzed during the present study are available from the corresponding author on reasonable request.

## Code availability
Data acquisition (Labview), analysis (Matlab), and simulation (Matlab) softwares used in this paper are described in Online Methods and will be available upon reasonable request.

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

## Acknowledgements

We thank M. Weck for the use of the clean room and T. Pinon and J. Palacci for help with the microfabrication process. This work was supported by a NIH grant R01DC005787 (A.R.). X-J.W. was supported by Naval Research grant N00014–17–1–2041, NIH grant R01MH062349, Science and Technology Commission of Shanghai Municipality grant 15JC1400104. J.B. was supported by a Human Frontier Science Program long-term postdoctoral fellowship (LT000132/2012) and by the Bettencourt Schueller Foundation.

## Author contributions

J.B., X.-J.W. and A.R. designed the project. J.B. performed the experiments and analyzed the results. J.B. and A.R. wrote the paper.

## Additional information

**Competing interests:** The authors declare no competing interests.

