## [Peer Review File · Nature Communications]

Editorial Note: This manuscript has been previously reviewed at another journal that is not operating a transparent peer review scheme. This document only contains reviewer comments and rebuttal letters for versions considered at Nature Communications . Mentions of prior referee reports have been redacted.

Reviewers' Comments:

Reviewer #1:

Remarks to the Author:

I have now read and considered carefully the revised manuscript and the responses to the reviewers. While I am delighted by the thoroughness of the responses, I am a little disappointed by the only moderate extra work that went into the manuscript itself, beyond the altered Introduction and Discussion. Nevertheless, I am convinced by many of the authors' arguments. I am now leaning towards publication of the manuscript in Nature Communications if the authors can accommodate my following final comment:

In their manuscript, but in particular in their response to reviewers 2 and 3, the authors seem to discard ALL previous models, regardless of whether these models contradict their results (those that tend to display synchrony) or concur with their results (those that tend to show rate propagation. Especially the rejection of the latter papers (vanRossum 2002, Litvak 2003, Vogels 2005, and, though not mentioned

Kumar, Rotter, Aertsen, JNeuro 2008), along with the authors reference to "unrealistic" or "extreme" conditions seems puzzling, especially because their various supplementary figures (see especially S4, S8, S10) show often striking similarities with some of these papers. The authors go on to postulate that all these models could be fixed by adding NMDA currents. This is believable, but thought-provoking. I have the following questions:

Can the authors directly compare their results to the mentioned models? I would be satisfied if the authors could adapt their non-embedded feed forward model from S1 to those of Diesmann 99 and van Rossum 02 (essentially the same model with varying neuron numbers, and degrees of connectivity, and some additional details) and show *convincingly* that these models fail to reproduce their results? Could the authors then go on to add the postulated NMDA currents to rescue the result? If the authors' claim is correct, this should be easy to show, and roughly 4 - 5 weeks work for a motivated researcher. Added bonus work would be repetition of the comparison, with (a) inclusion of inhibitory neurons in the FF structure, and (b) a demonstration of jitter modulated propagation in the model.

As with previous comments, the authors may want to claim that this is too much work, or outside of the scope of their work, but given that the model already exists, the addition would be a strong argument for their discard of previous models. The authors would establish a new standard model of propagation. Additionally, given the authors' track records, an explicit omission of such a model would raise some questions regarding the validity of their claims.

Reviewer #2:

Remarks to the Author:

We reviewed a previous version of this manuscript for [Redacted]. This manuscript appears to have been triaged to Nature Communications, with authors rebutting to our previous review. In their rebuttal to our queries, the authors have mostly argued and made very little substantive effort to address our points. I count a total of eight MINOR changes to the manuscript in responses to our six MAJOR points (which the authors in their rebuttal for some unclear reason downgrade to "concerns" rather than major points). This is not sufficient to satisfy us.

The responses are furthermore quite dismissive, for example stating "Actually demonstrating that these occur in vivo is difficult." That it is difficult to show does not mean you can still make those

claims. There is a striking lack of concrete and direct evidence in the rebuttal.

Since the authors are unable to take our criticism seriously, I have to recommend rejection. I cannot see how this manuscript is suitable for a solid journal such as Nature Communications.

Additional points:

Concern 3 - Why don't the activity from recurrent connections and feedback from downstream layers not contribute to activity recorded in patched cells? If the connections formed in this culture are bidirectional and symmetrical, why would recurrent connections or "feedback" connections act differently than feedforward, where feedforward connections are transmitting the "rate code", but feedback is minimal and can just be ignored. On the same note, why would NMDA blockade only affect feedforward, whereas responses from recurrent connections remained (line 496)

Line 73-74 "the results reflect general operating principals independent of detailed organization or cell types" Maybe the authors should say something like "the results reflect general operating principals of cultured systems..." instead.

In general, we think the authors should tone down the assertive language used in the discussion (e.g. line 500).

Reviewer #3:

Remarks to the Author:

The authors show that propagation of activity in a network of mouse primary cortical neurons depends on network density, such that timing information is preserved in sparse networks, but only rate information is propagated in denser networks, and this rate propagation requires NMDA receptors. These results will be of interest to specialists in the field, primarily for the advanced data analysis employed.

The manuscript has been adequately revised in most parts, but there are still some issues I think the authors should consider:

1. The authors appear to use the term 'temporal coding' with two different meanings, first, information encoded in the precise timing of spikes ('synfire mode'), and, second, information encoded in the timing precision of spikes ('jitter'). This may be confusing for the reader unless this distinction is made explicitly clear at the outset.

2. The authors insist their results stand in contrast to theoretical predictions. However, the cited predictions were made for simple feedforward networks, not for (recurrent) networks of cultured neurons; thus, the predictions and experimental results are not directly related. Having said that, some readers might even say that the activity propagation in sparse networks illustrated in Figure 2a is not dissimilar to the 'synfire chain' mode of propagation that theory predicts for feedforward networks, thus supporting the suggestion that this mode of propagation might be biologically relevant, contrary to the conclusion reached by the authors.

3. The authors should improve the precision of biological terminology. For example, they use the terms 'AMPA and NMDA synapses', which in the cultures are probably not different synapses, but rather one type of synapse using glutamate as neurotransmitter, but with synaptic events mediated by two types of receptor: AMPA receptors (not tested) and NMDA receptors.

Response to referee #1

Remarks to the Author

I have now read and considered carefully the revised manuscript and the responses to the reviewers. While I am delighted by the thoroughness of the responses, I am a little disappointed by the only moderate extra work that went into the manuscript itself, beyond the altered Introduction and Discussion. Nevertheless, I am convinced by many of the authors' arguments. I am now leaning towards publication of the manuscript in Nature Communications if the authors can accommodate my following final comment:

In their manuscript, but in particular in their response to reviewers 2 and 3, the authors seem to discard ALL previous models, regardless of whether these models contradict their results (those that tend to display synchrony) or concur with their results (those that tend to show rate propagation. Especially the rejection of the latter papers (vanRossum 2002, Litvak 2003, Vogels 2005, and, though not mentioned Kumar, Rotter, Aertsen, JNeuro 2008), along with the authors reference to "unrealistic" or "extreme" conditions seems puzzling, especially because their various supplementary figures (see especially S4, S8, S10) show often striking similarities with some of these papers. The authors go on to postulate that all these models could be fixed by adding NMDA currents. This is believable, but thought-provoking. I have the following questions:

*Can the authors directly compare their results to the mentioned models? I would be satisfied if the authors could adapt their non-embedded feed forward model from S1 to those of Diesmann 99 and van Rossum 02 (essentially the same model with varying neuron numbers, and degrees of connectivity, and some additional details) and show *convincingly* that these models fail to reproduce their results? Could the authors then go on to add the postulated NMDA currents to rescue the result? If the authors' claim is correct, this should be easy to show, and roughly 4 - 5 weeks work for a motivated researcher. Added bonus work would be repetition of the comparison, with (a) inclusion of inhibitory neurons in the FF structure, and (b) a demonstration of jitter modulated propagation in the model.*

As with previous comments, the authors may want to claim that this is too much work, or outside of the scope of their work, but given that the model already exists, the addition would be a strong argument for their discard of previous models. The authors would establish a new standard model of propagation. Additionally, given the authors' track records, an explicit omission of such a model would raise some questions regarding the validity of their claims.

Our response:

We thank the reviewer for his/her careful reading of the manuscript. As suggested, we now include simulations of the multilayer network. The description of the model is included in the Methods section. Intrinsic parameters and connectivity are directly derived from our experimental measurements. With these parameters, a sufficiently dense feedforward network ($N \gtrsim 500$ neurons per layer) can propagate a pulse packet similarly to the (Diesmann *et al.*, 1999) paper (Supplementary Fig. 18, left column). Adding recurrent excitatory connections prolonged the response (Supplementary Fig. 18, second column from

left). However, with our parameters, adding the inhibitory neurons prevented the propagation from occurring even in the presence of recurrent excitatory connections (Supplementary Fig. 18, third column from left). This is because inhibition is strong owing to a high connection probability between inhibitory to excitatory neurons and to the large amplitude IPSPs, which as we mention in the text, scaled with network size. Adding an NMDA component to excitatory synaptic current could rescue propagation but with a different activity profile that displayed a persistent phase (Supplementary Fig. 18, right column). Surely, propagation could also be possible without the NMDA current with other model parameters. But here, we used parameters derived from experiments and did not tune them.

Our simulated networks are able to reproduce the main features of experimental results:

- We observed 2 modes of propagation depending on the number of neurons: a “synfire mode” that propagated a short transient activity but only over 2-3 layers and a “rate mode” in which the transient was rapidly replaced by sustained activity. The transition in the mode of propagation was also accompanied by a transition in probability of propagation and by an increase in propagation delay as seen in experiments (cf Fig. 2 of the main text and Supplementary Fig. 3)
- Jitter in the stimulus can modulate the firing rate and therefore information about stimulus jitter can be propagated in dense networks (cf Fig. 4 of the main text and Supplementary Fig. 11).
- The firing rate in layers 2 to 5 can be used to read out the firing rate in layer 1 (cf Fig. 5 of the main text and Supplementary Fig. 13).
- This mode of propagation depends on NMDA current and recurrent connectivity within layers. Pharmacological perturbations could be reproduced with simulations (cf Fig. 7 of the main text and Supplementary Fig. 18). We could also use simulations to characterize the contributions of recurrent and feedback connections: whereas recurrent connections are necessary to transmit the input via NMDA-mediated currents, removing feedback connections does not prevent propagation in our simulations (Supplementary Fig. 19).

Response to referee #2

Remarks to the Author

We reviewed a previous version of this manuscript for [Redacted]. This manuscript appears to have been triaged to Nature Communications, with authors rebutting to our previous review. In their rebuttal to our queries, the authors have mostly argued and made very little substantive effort to address our points. I count a total of eight MINOR changes to the manuscript in responses to our six MAJOR points (which the authors in their rebuttal for some unclear reason downgrade to "concerns" rather than major points). This is not sufficient to satisfy us. The responses are furthermore quite dismissive, for example stating "Actually demonstrating that these occur in vivo is difficult." That it is difficult to show does not mean you can still make those claims. There is a striking lack of concrete and direct evidence in the rebuttal.

Since the authors are unable to take our criticism seriously, I have to recommend rejection. I cannot see how this manuscript is suitable for a solid journal such as Nature Communications.

Additional points:

Concern 1: *in relation to Concern 3 from the previous round- Why don't the activity from recurrent connections and feedback from downstream layers not contribute to activity recorded in patched cells? If the connections formed in this culture are bidirectional and symmetrical, why would recurrent connections or "feedback" connections act differently than feedforward, where feedforward connections are transmitting the "rate code", but feedback is minimal and can just be ignored. On the same note, why would NMDA blockade only affect feedforward, whereas responses from recurrent connections remained (line 496).*

Our response: Without providing a definitive answer about the role of recurrent and feedback connections, we point out that there are much more recurrent connections than feedback connections. This is because the connection probability decreases as a function of distance with a Gaussian profile (see Supplementary Fig. 2). Because of the large distance across layers, the connections are sparser than those within layer. Therefore, it is likely that recurrent connections play a stronger role in propagation than feedback connections. Unfortunately, we cannot directly measure the relative contributions of these two types of connections with experiments. However, simulations of neuronal networks offer a way to gain insight about the underlying mechanism. We now address this issue with simulations of multilayer networks with parameters derived from experimental measurements. The simulations reproduced the main features of experiments (see response to reviewer #1) and show that whereas recurrent connections are necessary to transmit the input via NMDA-mediated currents, removing feedback connections does not prevent propagation in our simulations (Supplementary Fig. 19). We added the following paragraph in the discussion to emphasize this point (lines 498-501):

We used simulations to further understand the impact of the different connection types (Supplementary Fig. 19). In purely feedforward networks made of excitatory and inhibitory neurons; propagation failed after few layers. Whereas the addition of feedback connections to the network did not qualitatively change the behavior, the presence of recurrent connections strongly supported propagation.

Concern 2: *Line 73-74 "the results reflect general operating principals independent of detailed organization or cell types" Maybe the authors should say something like "the results reflect general operating principals of cultured systems..." instead.*

Our response: We refer here to published work where we extensively described the spontaneous and evoked dynamics of cortical cultures (Barral & Reyes, Nature Neuroscience 2016). In this study, we compared our results in cortical cultures to *in vivo* brain dynamics and to results obtained from simulations. The dynamics of the culture networks recapitulated several aspects of the *in vivo* dynamics (balance between excitation and inhibition, rate-correlation relationship, active decorrelation, decreased Fano factors during evoked firing) without tuning the network. We are thus confident that our results are

not restricted to cultures and instead reflects emergent properties of networks in general. Thus, we would prefer not to include the suggested qualifying statement.

Concern 3: *In general, we think the authors should tone down the assertive language used in the discussion (e.g. line 500).*

Our response: About line 500 of the manuscript (beginning of the 4th paragraph of the discussion), we are simply stating that in the case of a firing rate coding scheme, the information capacity of a neuron (i.e. the number of levels that a neuron is able to encode) is increased if the neuron can generate a larger range of firing rate. In other words, we are simply saying that if there is a fixed amount of noise in the firing rate, a neuron can encode more levels of firing rates if it can explore a larger range of firing activity. Our statement might sound assertive because it is a very general relation derived from the Shannon theory of information (Shannon, A mathematical theory of communication. Bell Syst. Tech. 1948). The Shannon formula tells that the maximum rate at which information can be transmitted is function of signal-to-noise ratio P/N: $C=1/2*\log_2(1+P/N)$. Therefore, if the signal P (i.e. the range of firing rate that the neuron can explore) is increased, the channel capacity is increased. The sentence that the referee mentions doesn't refer to the culture system in particular but is very general. We therefore rewrote this sentence to avoid any misinterpretation (lines 502-504):

In the framework of information theory, the information capacity of a neuron is related to the range of firing that the neuron can explore. Because noise interferes with the discrimination of nearby levels of firing activity, the capability to generate a broad range of firing is important for representing information as rate: networks that have a narrow range of responses to a wide range of stimuli, for example, have limited information capacity.

The referee might also be objecting to our statement that temporal code is converted to a rate code *in vivo*, as it was in our culture experiments (lines 508-511 of the original manuscript). If this is a concern, we could also remove the passage (Lines 511-514): *“The transformation of temporal code to a rate code and subsequent propagation through layers is of some functional significance: for example, phase-locked, temporally precise signals generated by specialized brainstem neurons are lost en route to cortex (Ahissar et al., 2001; Ahissar et al., 2000; Gao et al., 2016; Gao and Wehr, 2015; Lu et al., 2001).”*

Response to referee #3

Remarks to the Author

The authors show that propagation of activity in a network of mouse primary cortical neurons depends on network density, such that timing information is preserved in sparse networks, but only rate information is propagated in denser networks, and this rate propagation requires NMDA receptors. These results will be of interest to specialists in the field, primarily for the advanced data analysis employed. The

manuscript has been adequately revised in most parts, but there are still some issues I think the authors should consider.

Concern 1: *The authors appear to use the term ‘temporal coding’ with two different meanings, first, information encoded in the precise timing of spikes (‘synfire mode’), and, second, information encoded in the timing precision of spikes (‘jitter’). This may be confusing for the reader unless this distinction is made explicitly clear at the outset.*

Our response: We agree with the reviewer that “temporal information” cannot refer to two distinct features, namely information about the spike timing and information about the stimulus jitter. To conform to the most commonly used definition, we kept “temporal information” to refer to information contained in the precise spike timing. We explicitly wrote information about stimulus jitter when relevant (e.g lines 70, 230, 277).

Concern 2: *The authors insist their results stand in contrast to theoretical predictions. However, the cited predictions were made for simple feedforward networks, not for (recurrent) networks of cultured neurons; thus, the predictions and experimental results are not directly related. Having said that, some readers might even say that the activity propagation in sparse networks illustrated in Figure 2a is not dissimilar to the ‘synfire chain’ mode of propagation that theory predicts for feedforward networks, thus supporting the suggestion that this mode of propagation might be biologically relevant, contrary to the conclusion reached by the authors.*

Our response: The theoretical papers that we cite are based on either purely feedforward networks (e.g. Diesman et al., 2000; van Rossum et al., 2002) or feedforward chain embedded in a recurrent network (e.g. Aviel et al., 2003; Kumar et al., 2008; Mehring et al., 2003; Vogels and Abbott, 2005). The later model is probably a better description of our biological network. To directly relate our experiments with theory, we now add simulations of multilayer network whose parameters are directly extracted from experimental measurements (see response to reviewer #1 and the modified manuscript).

As the reviewer points out, the activity propagation in Fig. 2a is very similar to the synfire chain and we actually named this mode of propagation as “synfire mode”. What was not predicted by theory is the other mode of propagation (“rate mode”) that we show in Fig. 2b and that we observed in dense networks. In this case, recurrent activity and NMDA-based currents need to be included in the model to explain the results. Using simulations, we now show that recurrent activity and NMDA currents are both necessary to transmit activity in the rate mode (Supplementary Figs. 18 and 19).

Concern 3: *The authors should improve the precision of biological terminology. For example, they use the terms ‘AMPA and NMDA synapses’, which in the cultures are probably not different synapses, but rather one type of synapse using glutamate as neurotransmitter, but with synaptic events mediated by two types of receptor: AMPA receptors (not tested) and NMDA receptors.*

Our response: Instead of “AMPA and NMADA synapses”, we now write instead “AMPA and NMDA-mediated components”.

Reviewers' Comments:

Reviewer #1:

Remarks to the Author:

I have now had the opportunity to review the revised manuscript, and I have little to add in terms of criticism regarding the provided materials. I still believe that the additional modelling work deserves a more prominent spot in the manuscript, as well as a further and more comprehensive analysis, and a good discussion regarding the implications of the results for possible neural codes, because it would strengthen the standing and rank of their results. Ultimately, it is up to the authors to keep their paper in this state, but it's a shame.

I join the other reviewers in bemoaning that only the very minimum possible amount of work for every suggestion has been performed, and it is clear from the tone and attitude of the last two rebuttals that there is no willingness to entertain any suggestions that are outside of the immediate intellectual comfort zone of the investigators.

This reviewer continues to be flabbergasted why so few of the suggestions by the three people who will have interacted with the manuscript most have been treated in good faith, and I have only seldomly had a more antagonistic experience with a manuscript that is ultimately solid work, but could have been much, much better.

REVIEWERS' COMMENTS:

Reviewer #1 (Remarks to the Author):

I have now had the opportunity to review the revised manuscript, and I have little to add in terms of criticism regarding the provided materials. I still believe that the additional modelling work deserves a more prominent spot in the manuscript, as well as a further and more comprehensive analysis, and a good discussion regarding the implications of the results for possible neural codes, because it would strengthen the standing and rank of their results. Ultimately, it is up to the authors to keep their paper in this state, but it's a shame.

I join the other reviewers in bemoaning that only the very minimum possible amount of work for every suggestion has been performed, and it is clear from the tone and attitude of the last two rebuttals that there is no willingness to entertain any suggestions that are outside of the immediate intellectual comfort zone of the investigators.

This reviewer continues to be flabbergasted why so few of the suggestions by the three people who will have interacted with the manuscript most have been treated in good faith, and I have only seldomly had a more antagonistic experience with a manuscript that is ultimately solid work, but could have been much, much better.

Our response: The manuscript has evolved extensively from the first submission because we took very seriously the reviewers' comments. We added new analysis, new experiments and developed a completely new network model to fully describe propagation in our multilayer networks. In the last version, we added 5 supplementary figures related to the simulations and we feel that adding more details would simply cloud the issue. The study is primarily experimental in nature and that although the computational work strengthens the interpretation of the results, they do not affect substantially the conclusions. We used pharmacological perturbation and recorded spikes and subthreshold membrane potential fluctuations to fully describe how activity propagates in the networks. Simulations reproduced the results quite accurately. As suggested by the referee, we now added these computational results to the main text (new figure 8).

A discussion about potential neural codes was included in the manuscript. We were reluctant to say more because our attempt to relate the results to coding *in vivo* in the original submission was heavily criticized by one of the referees. We were therefore very careful not to stray too far from the data or to speculate too much.

The reviewers' comments and our responses are likely to reflect the debate that will occur in the general neuroscience community. We have therefore agreed to let Nature Communications publish reviewers' comments and our responses along with the manuscript.